# The target of rapamycin signaling pathway regulates vegetative development, aflatoxin biosynthesis, and pathogenicity in *Aspergillus flavus*

**Guoqi Li, Xiaohong Cao, Elisabeth Tumukunde, Qianhua Zeng, Shihua Wang\***

State Key Laboratory of Ecological Pest Control for Fujian and Taiwan Crops, Key Laboratory of Pathogenic, Fungi and Mycotoxins of Fujian Province, School of Life Sciences, Fujian Agriculture and Forestry University, Fuzhou, China

**\*For correspondence:**
wshyyl@sina.com

**Competing interest:** The authors declare that no competing interests exist.

**Abstract** The target of rapamycin (TOR) signaling pathway is highly conserved and plays a crucial role in diverse biological processes in eukaryotes. Despite its significance, the underlying mechanism of the TOR pathway in *Aspergillus flavus* remains elusive. In this study, we comprehensively analyzed the TOR signaling pathway in *A. flavus* by identifying and characterizing nine genes that encode distinct components of this pathway. The FK506-binding protein Fkbp3 and its lysine succinylation are important for aflatoxin production and rapamycin resistance. The TorA kinase plays a pivotal role in the regulation of growth, spore production, aflatoxin biosynthesis, and responses to rapamycin and cell membrane stress. As a significant downstream effector molecule of the TorA kinase, the Sch9 kinase regulates aflatoxin $B_1$ ($AFB_1$) synthesis, osmotic and calcium stress response in *A. flavus*, and this regulation is mediated through its S_TKc, S_TK_X domains, and the ATP-binding site at K340. We also showed that the Sch9 kinase may have a regulatory impact on the high osmolarity glycerol (HOG) signaling pathway. TapA and TipA, the other downstream components of the TorA kinase, play a significant role in regulating cell wall stress response in *A. flavus*. Moreover, the members of the TapA-phosphatase complexes, SitA and Ppg1, are important for various biological processes in *A. flavus*, including vegetative growth, sclerotia formation, $AFB_1$ biosynthesis, and pathogenicity. We also demonstrated that SitA and Ppg1 are involved in regulating lipid droplets (LDs) biogenesis and cell wall integrity (CWI) signaling pathways. In addition, another phosphatase complex, Nem1/Spo7, plays critical roles in hyphal development, conidiation, aflatoxin production, and LDs biogenesis. Collectively, our study has provided important insight into the regulatory network of the TOR signaling pathway and has elucidated the underlying molecular mechanisms of aflatoxin biosynthesis in *A. flavus*.

## eLife assessment

This manuscript provides **important** information about the influence of TOR signaling pathway on development and aflatoxin production in the plant and human fungal pathogen Aspergillus flavus. Compared to an earlier version, the authors have addressed most of the concerns of the reviewers, including the **convincing** demonstration of the essential TOR pathway in this fungus by constructing a xylose promoter mutant strain.

## Introduction

*Aspergillus flavus* is a highly significant phytopathogenic fungus that frequently contaminates a diverse array of agricultural crops, leading to substantial economic losses in the food and agriculture industry and posing considerable health risks, particularly in developing countries (*Klich, 2007*). Moreover, *A. flavus* is recognized as an opportunistic human pathogenic fungus that can cause aspergillosis in immune-compromised patients (*Hedayati et al., 2007*). The fungus is notorious for its production of aflatoxins, which are potent secondary metabolites with toxic and carcinogenic properties and are classified among the most powerful carcinogens known in nature, with established links to liver cancer in both humans and animals (*Khan et al., 2021*). Aflatoxin biosynthesis is a multifaceted and intricate biochemical process encompassing a cascade of enzymatic reactions. Although the majority of the enzyme reactions and genes involved in aflatoxin biosynthesis have been elucidated (*Yabe and Nakajima, 2004*), the related signaling pathway network and specific regulatory mechanisms governing aflatoxin biosynthesis remain elusive. Therefore, exploring the regulatory mechanism of the aflatoxin biosynthesis signaling pathway would provide new insights for preventing *A. flavus* and aflatoxin contamination.

In eukaryotes, the target of rapamycin (TOR) signaling pathway is highly conserved and plays essential roles in various significant biological processes, including ribosome biosynthesis, cell growth, and autophagy (*Wullschleger et al., 2006*). The serine/threonine-protein kinase Tor serves as a crucial protein component in the TOR signaling pathway, which interacts with other proteins to form multi-protein complexes (*De Virgilio and Loewith, 2006*). In *Saccharomyces cerevisiae*, TOR exists in two distinct multiprotein functional complexes, namely target rapamycin complex 1 (TORC1) and TORC2 (*Loewith et al., 2002*). TORC1-mediated signaling governs cellular growth by modulating various growth-related mechanisms and exhibits sensitivity to rapamycin. Conversely, TORC2-mediated signaling primarily regulates cytoskeletal remodeling but remains unaffected by rapamycin (*Loewith et al., 2002*). In yeast, there are two *tor* genes, *tor1* and *tor2*, whereas in higher eukaryotes such as plants, animals, and filamentous fungi, there is only one *tor* gene (*Beauchamp and Platanias, 2013*; *Fu et al., 2021*; *Teichert et al., 2006*).

In the fungal kingdom, the TOR signaling pathway has been extensively investigated in three prominent species: the budding yeast *S. cerevisiae* (*Crespo and Hall, 2002*; *Rohde et al., 2008*), the fission yeast *Schizosaccharomyces pombe* (*Ishiguro et al., 2013*; *Otsubo and Yamamato, 2008*; *Petersen, 2009*), and the human pathogen *Candida albicans* (*Bastidas et al., 2009*; *Liu et al., 2017*). In *S. cerevisiae*, the macrolide antibiotic rapamycin forms a complex with the cytosolic protein Fkbp12, which effectively inhibits the catalytic function of the Tor kinase by interacting with its FRB domain (*Hu et al., 2016*). The Tap42 phosphatase complex serves as the primary target of the Tor kinase and plays a crucial role in the regulation of downstream effectors. Additionally, the Tap42 protein interacts with the Tip41 protein to collaboratively modulate the activities of phosphatases, including PP2A and Sit4 (*Wang et al., 2003*; *Yan et al., 2006*). Among these, the Sit4 phosphatase is known to dephosphorylate the transcription factor Gln3, thereby regulating the nitrogen source metabolism pathway (*Tate et al., 2019*). In addition, the AGC kinase Sch9, an immediate effector of TORC1, is involved in regulating various cellular function processes, such as ribosome biosynthesis, translation initiation, and entry into the $G_0$ phase (*Urban et al., 2007*). The TOR signaling pathway has been documented in various filamentous fungi, including *Fusarium graminearum* (*Yu et al., 2014*; *Liu et al., 2019*), *Magnaporthe oryzae* (*Sun et al., 2018*), *Phanerochaete chrysosporium* (*Nguyen et al., 2020*), *Podospora anserine* (*Dementhon et al., 2003*), *Aspergillus nidulans* (*Fitzgibbon et al., 2005*), *Aspergillus fumigatus* (*Alves de Castro and Dos Reis, 2016*; *Baldin et al., 2015*), *Fusarium fujikuroi*, *Botrytis cinerea* (*Meléndez et al., 2009*), *Fusarium oxysporum* (*López-Berges et al., 2010*), and *Mucor circinelloides* (*Bastidas et al., 2012*). However, the TOR signaling pathway in *A. flavus* remains unexplored.

In eukaryotic cells, the TOR signaling pathway and the mitogen-activated protein kinase (MAPK) signaling pathways play an important role in regulating adaptive responses to extra- and intracellular conditions (*Madrid et al., 2016*). Several studies have demonstrated that the TOR signaling pathway interacts with various other signaling pathways, such as MAPK and cell wall integrity (CWI) pathway. These pathways are recognized to participate in crosstalk, forming a complex metabolic network that regulates cellular processes (*Torres et al., 2002*; *Chen et al., 2019*; *Gu et al., 2015*; *Qian et al., 2018*; *Qian et al., 2021*). However, the underlying mechanisms of multiple crosstalks between the TOR and

other signaling pathways in *A. flavus* remain largely unknown. Therefore, we intended to identify the genes associated with the TOR signaling pathway and elucidate their roles and contributions in regulating vegetative development and aflatoxin biosynthesis in *A. flavus*. Our results demonstrated that the TOR signaling pathway is involved in multiple cellular processes in *A. flavus*. Notably, our findings revealed a complex interplay between the TOR pathway and other signaling pathways (such as high osmolarity glycerol [HOG] and CWI) in *A. flavus*, which play a crucial role in regulating cellular growth and survival under environmental stress conditions.

## Results

### Rapamycin exhibits inhibitory effects on the growth, sporulation, sclerotia formation, and aflatoxin production in *A. flavus*

Rapamycin, a secondary metabolite synthesized by *Streptomyces hygroscopicus*, exhibits efficacy in suppressing specific filamentous fungi. In *F. graminearum*, rapamycin has a significant inhibitory impact on growth and asexual reproduction (*Yu et al., 2014*). To investigate the effects of rapamycin on vegetative development, sclerotia formation, and secondary metabolism in *A. flavus*, we conducted an assay to determine the sensitivity of the wild-type (WT) strain to rapamycin. We observed that the WT strain exhibited a high sensitivity to rapamycin. We found that 100 ng/mL of rapamycin significantly inhibited mycelial growth and conidia formation (*Figure 1A, D, and E*). More importantly, the WT strain exhibited a significant reduction in sclerotia formation and aflatoxin synthesis (*Figure 1B, C, F, and G*). Additionally, rapamycin resulted in a significant decrease in spore germination rate and sparser conidiophores (*Figure 1—figure supplement 1A and B*). These results illustrated that the TOR signaling pathway may be involved in regulating the growth, development, and secondary metabolism of *A. flavus*. Calcofluor white (CFW) is a chemifluorescent blue dye commonly employed for its nonspecific binding affinity to fungal chitin. Fluorescence microscopy analysis revealed a notable reduction in the blue fluorescence on the cell wall following treatment with rapamycin (*Figure 1—figure supplement 1C*). The intracellular lipid droplets (LDs) within the hyphae were labeled with BODIPY. We observed that the hyphae treated with rapamycin contained more LDs compared to the control group (*Figure 1—figure supplement 1D*). These findings suggested that the TOR pathway may play important roles in maintaining CWI and facilitating the biogenesis of LDs.

To investigate the potential regulatory role of the TOR pathway in modulating the sporulation, sclerotia formation, and aflatoxin biosynthesis in *A. flavus*, we examined the expression levels of conidia-related, sclerotia formation-related, and aflatoxin-related genes after 3, 6, and 9 hr of treatment with rapamycin. To determine the effects of rapamycin on conidiation, qRT-PCR was conducted to assess the transcript levels of two conidia-related genes, *abaA* and *brlA*. We also conducted qRT-PCR to evaluate the expression levels of three regulators (*nsdC*, *nsdD*, and *sclR*) involved in sclerotia development, and genes involved in AFB$_1$ biosynthesis, such as the regulatory gene *aflS*, as well as several structural genes (*aflC* and *aflQ*). The strains treated with rapamycin exhibited a significant reduction in the expression levels of genes related to sporulation, sclerotia formation, and aflatoxin synthesis (*Figure 1H, I, and J*). Overall, these results indicated that the TOR pathway plays a crucial role in regulating the growth, sporulation, sclerotia formation, and aflatoxin biosynthesis in *A. flavus* by modulating the expression of these genes.

### The impact of Fkbp3 and its lysine succinylation on AFB$_1$ biosynthesis and rapamycin resistance in *A. flavus*

FK506-binding proteins (Fkbps) belong to a highly conserved immunophilin family, which are involved in various cellular functions, such as protein folding, immunosuppression, signaling transduction, and transcription (*Tong and Jiang, 2015*). Fkbps function as molecular switches by binding to target proteins and inducing conformational changes (*Tong and Jiang, 2015*). In *S. cerevisiae*, Fkbp12 and rapamycin interact to form a complex that exerts a negative regulatory effect on the activity of the Tor kinase (*Heitman et al., 1991*). The *fkbp12* gene homolog *fprA* in *A. nidulans* shares high levels of homology with that from *S. cerevisiae* (*Fitzgibbon et al., 2005*). According to the protein sequence of FprA from *A. nidulans*, four genes encoding putative proteins with FK506-binding domain, named *fkbp1* (AFLA_126880), *fkbp2* (AFLA_087480), *fkbp3* (AFLA_010910), and *fkbp4* (AFLA_128200), have been identified from the *A. flavus* genome. We generated four *fkbp* knockout strains using

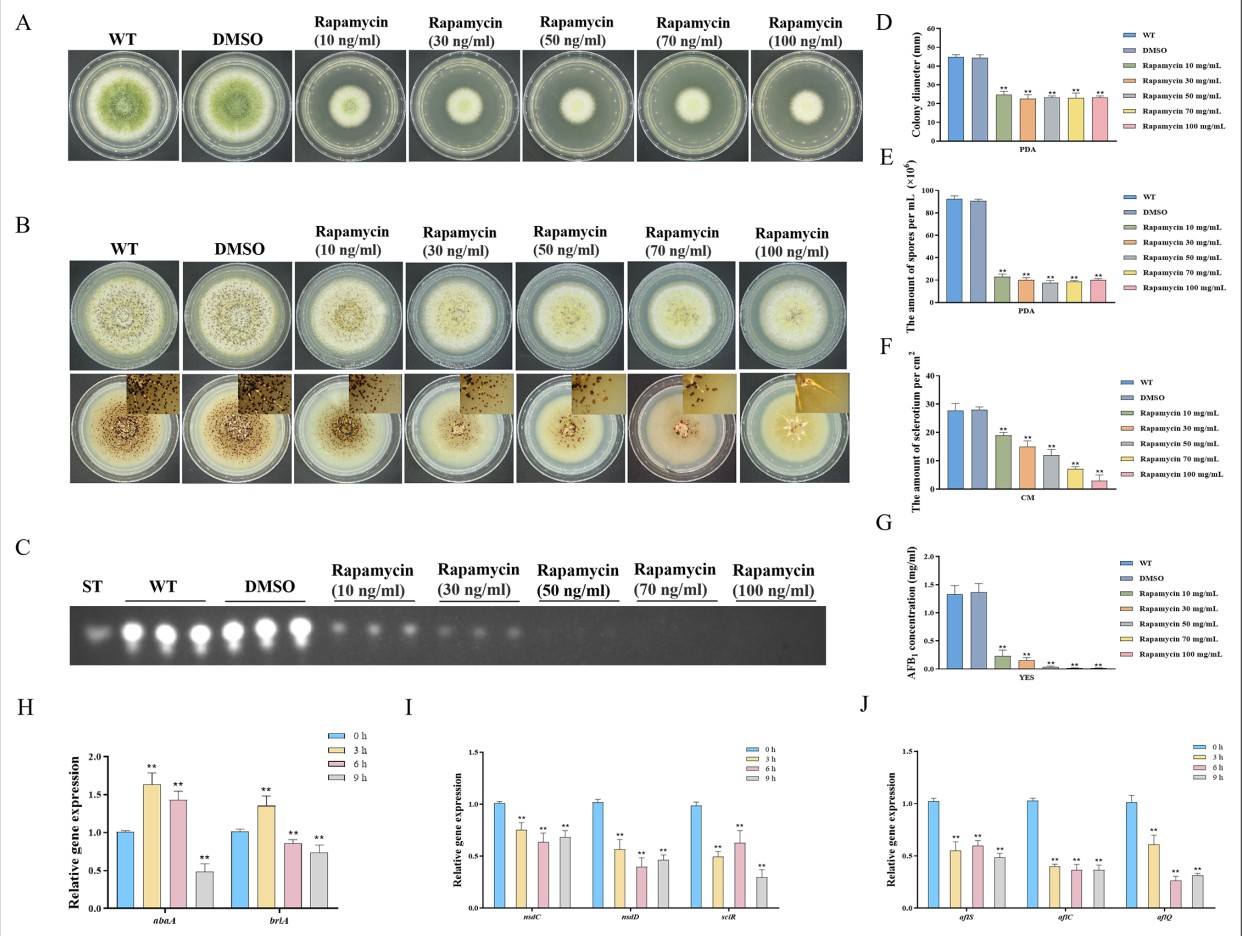

**Figure 1.** Impacts of rapamycin on the growth, sporulation, sclerotia formation, and aflatoxin production of *A. flavus*. (**A**) Colony morphology of the wild-type (WT) strain cultured on potato dextrose agar (PDA) medium amended with different concentrations of rapamycin at 37°C for 5 days. (**B**) Colony phenotype of the WT strain grown on CM medium amended with different concentrations of rapamycin at 37°C for 7 days. (**C**) Thin layer chromatography (TLC) assay of aflatoxin $B_1$ ($AFB_1$) production by the WT strain cultured in yeast extract-sucrose agar (YES) liquid media amended with different concentrations of rapamycin at 29°C for 6 days. (**D**) Statistical analysis of the colony diameter by the WT strain treated with different concentrations of rapamycin described in (**A**). (**E**) Conidial quantification of the WT strain treated with different concentrations of rapamycin as mentioned in (**A**). (**F**) Quantitative analysis of sclerotium formation by the WT strain treated with different concentrations of rapamycin described in (**B**). (**G**) Relative quantification of $AFB_1$ production by the WT strain treated with different concentrations of rapamycin as mentioned in (**C**). (**H**) Relative expression levels of conidia-related genes in WT strain treated with 100 ng/mL rapamycin after 3, 6, 9 hr. (**I**) Relative expression levels of sclerotia-related genes in WT strain treated with 100 ng/mL rapamycin after 3, 6, 9 hr. (**J**) Relative expression levels of aflatoxin biosynthesis regulatory and structural genes in WT strain treated with 100 ng/mL rapamycin after 3, 6, 9 hr. ** indicates that the significance level was p≤0.01, based on one-way ANOVA test with three replicates(n=3). Error bars represent the standard error of the mean (SEM).

The online version of this article includes the following figure supplement(s) for figure 1:

**Figure supplement 1.** Impacts of rapamycin on spore germination, conidiophore formation, and lipid droplet biogenesis of *A. flavus*.

homologous recombination, and all *fkbp* disruption strains were confirmed by PCR and sequencing analysis (*Figure 2—figure supplement 1A*). Phenotypic analyses showed that the mycelial growth and conidia formation of all the *fkbp* mutants were similar to the WT strain on the potato dextrose agar (PDA) medium (*Figure 2A*), suggesting that Fkbps have minimal influence on the process of vegetative development. We evaluated $AFB_1$ synthesis utilizing thin layer chromatography (TLC), and the findings demonstrated a noteworthy reduction in $AFB_1$ synthesis upon deletion of Fkbp3, in comparison to the WT strain (*Figure 2B and D*). Additionally, the quantity of sclerotia exhibited a significant decrease in all *fkbp* mutants compared to the WT strain (*Figure 2—figure supplement 2A and B*). The aforementioned data indicated that Fkbp3 plays a crucial role in sclerotia formation and aflatoxin biosynthesis in *A. flavus*.

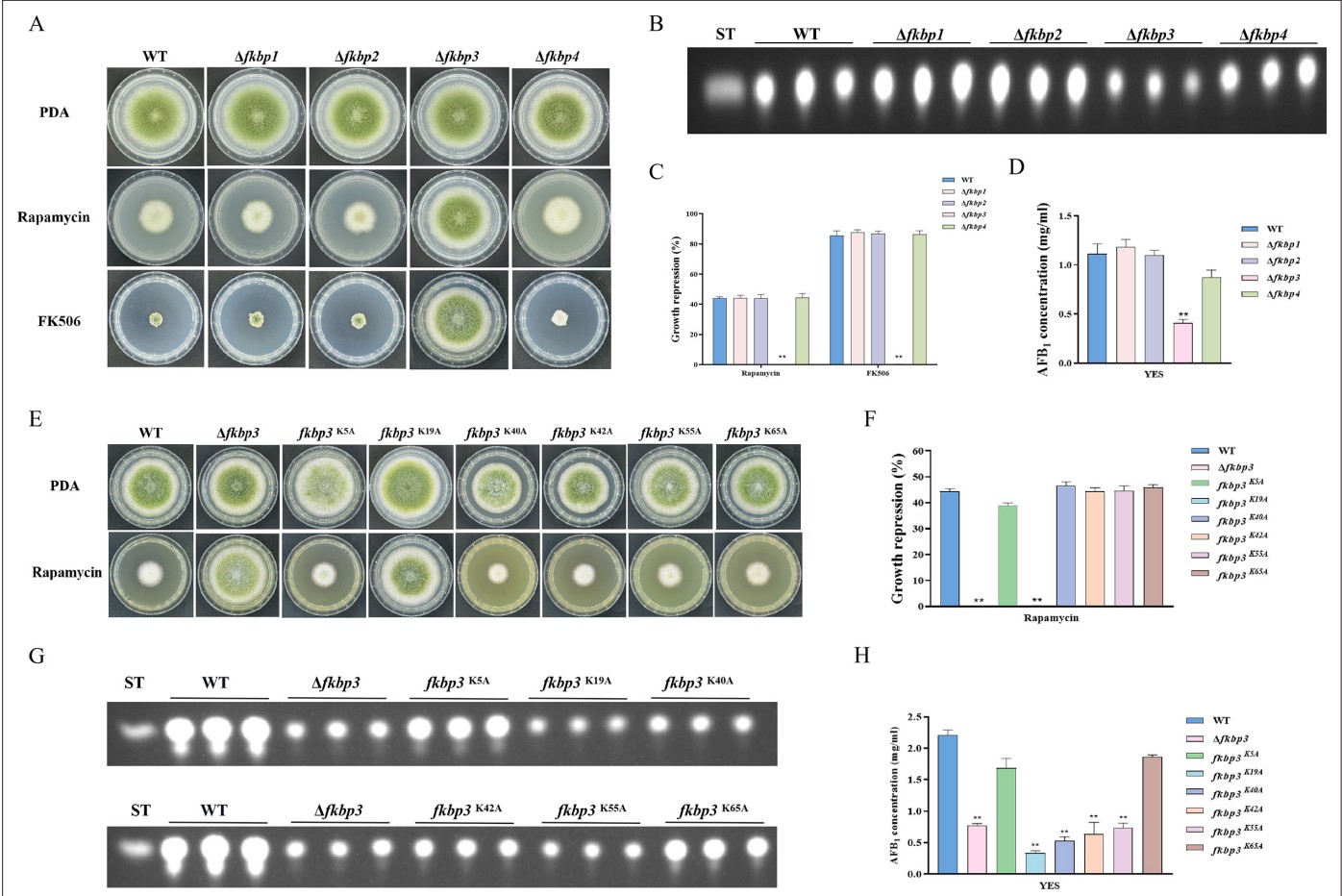

**Figure 2.** Disruption of *fkbp3* significantly increases the resistance of *A. flavus* to rapamycin and FK506. (**A**) Phenotype of the wild-type (WT) and all mutant strains (Δ*fkbp1*, Δ*fkbp2*, Δ*fkbp3*, and Δ*fkbp4*) grown on potato dextrose agar (PDA) amended with rapamycin and FK506 at 37°C for 5 days. (**B**) Thin layer chromatography (TLC) analysis of aflatoxin B$_1$ (AFB$_1$) production by the WT and all mutant strains (Δ*fkbp1*, Δ*fkbp2*, Δ*fkbp3*, and Δ*fkbp4*) cultured in yeast extract-sucrose agar (YES) liquid medium at 29°C for 6 days. (**C**) The growth inhibition rate of the WT and all mutant strains (Δ*fkbp1*, Δ*fkbp2*, Δ*fkbp3*, and Δ*fkbp4*) under rapamycin and FK506 stress. (**D**) AFB$_1$ quantitative analysis of the WT and all mutant strains (Δ*fkbp1*, Δ*fkbp2*, Δ*fkbp3*, and Δ*fkbp4*) as described in (**B**). (**E**) Phenotype of the WT and all mutant strains (Δ*fkbp3*, K5A, K19A, K40A, K42A, K55A, and K65A) grown on PDA amended with 100 ng/mL rapamycin at 37°C for 5 days. (**F**) The growth inhibition rate of the WT and all mutant strains (Δ*fkbp3*, K5A, K19A, K40A, K42A, K55A, and K65A) under rapamycin stress. (**G**) TLC assay of AFB$_1$ production by the WT and all mutant strains (Δ*fkbp3*, K5A, K19A, K40A, K42A, K55A, and K65A) cultured in YES liquid medium at 29°C for 6 days. (**H**) Relative quantification of AFB$_1$ production in the WT and all mutant strains (Δ*fkbp3*, K5A, K19A, K40A, K42A, K55A, and K65A) as mentioned in (**G**). ** indicates that the significance level was $p \le 0.01$, based on one-way ANOVA test with three replicates(n=3). Error bars represent the standard error of the mean (SEM).

The online version of this article includes the following source data and figure supplement(s) for figure 2:

**Figure supplement 1.** Construction of all mutants using homologous recombination.

**Figure supplement 1—source data 1.** Source data for DNA gel electrophoresis: original and annotated images for *Figure 2—figure supplement 1*.

**Figure supplement 2.** Fkbps regulate sclerotia biosynthesis in *A. flavus*.

To determine the specific Fkbp responsible for rapamycin sensitivity, all mutant strains were cultured on PDA solid plates amended with 100 ng/mL rapamycin. The strain with disrupted *fkbp3* showed significant resistance to rapamycin, while the deletion mutants of *fkbp1*, *fkbp2*, and *fkbp4* did not exhibit any alteration in resistance to rapamycin (***Figure 2A and C***). These results demonstrated that Fkbp3 serves as the principal target for rapamycin in *A. flavus*. In the FK506 sensitivity analysis, we observed a significant suppression of mycelial growth and conidia formation in WT strain with the addition of 10 ng/mL FK506 to the PDA medium. Additionally, the deletion of *fkbp3* led to the heightened resistance to FK506 compared to the WT strain (***Figure 2A and C***). These findings suggested that Fkbp3 is mainly involved in the modulation of FK506 resistance in *A. flavus*.

Previous studies have shown that lysine succinylation plays a significant role in aflatoxin production and pathogenicity in *A. flavus* (**Ren et al., 2018**). Based on the analysis of our succinylome data, we have identified six reliable succinylation sites (K5, K19, K40, K42, K55, and K65) on the Fkbp3 protein. To identify the function of Fkbp3 succinylation sites, site-directed mutations were constructed according to the homologous recombination strategy. Compared to the WT strain, all point mutants exhibited a similar phenotype in vegetative growth and conidiation. Interestingly, we observed that the K19A mutation resulted in increased resistance to rapamycin (**Figure 2E and F**). Additionally, the K19A strain exhibited a significant reduction in the production of $AFB_1$ compared to the WT strain (**Figure 2G and H**). These findings suggested that the succinylation of Fkbp3 at K19 plays a crucial role in rapamycin resistance and aflatoxin biosynthesis. Furthermore, we found that various point mutants, including K40A, K42A, and K55A, exhibited reduced levels of $AFB_1$ production in comparison to the WT strain (**Figure 2G and H**), suggesting that the succinylation of Fkbp3 at other sites also contributes to $AFB_1$ biosynthesis in *A. flavus*. The above results indicated that the K19 site likely plays a more prominent role in Fkbp3 succinylation compared to other sites.

## The TorA kinase plays a critical role in *A. flavus*

In *S. cerevisiae*, the Tor kinase is known to interact with a diverse range of proteins, resulting in the formation of complex structures known as TORC1 and TORC2 (**Weisman, 2016**). TORC1 can detect various signals, including nutrients, growth factors, and environmental stress. It plays a crucial role in regulating cellular processes such as gene transcription, protein translation, ribosome synthesis, and autophagy (**Loewith and Hall, 2011**). To identify the ortholog of the Tor protein in *A. flavus*, the *S. cerevisiae* Tor protein was utilized as search reference in the *A. flavus* genome database using the Basic Local Alignment Search Tool (https://blast.ncbi.nlm.nih.gov/Blast.cgi). The genome of *A. flavus* contains a solitary ortholog of the *torA* gene (AFLA_044350), which encodes a protein exhibiting 48.91% similarity to *S. cerevisiae* Tor2. To elucidate the role of *torA* in *A. flavus*, we attempted to obtain *torA* deletion strains by homologous recombination, but not successful. As an alternative approach, a mutant strain with a xylose promoter mutation, referred as *xylPtorA*, was constructed to validate the function of *torA*. Subsequent confirmation via PCR analysis ensured the integrity of the mutant strain. Xylan and xylose are highly effective inducers, whereas glucose strongly represses the activity of the promoter. To ascertain the impact of *torA* on *A. flavus*, we conducted a comparative analysis of *torA* transcript levels between the *xylPtorA* and the WT strains. Our findings revealed that the *torA* transcript levels in the *xylPtorA* strain were markedly reduced compared to the WT strain (**Figure 3—figure supplement 1**). Both the WT and *xylPtorA* strains were cultured in yeast extract-xylose agar (YXT) medium supplemented with xylose and yeast extract-glucose agar (YGT) medium devoid of xylose for a duration of 5 days at 37°C under dark conditions. Remarkably, the *xylPtorA* strain exhibited complete inability to thrive in the absence of xylose, with only partial restoration of growth observed upon xylose supplementation (**Figure 3A and D**). Notably, the *xylPtorA* strain also demonstrated a markedly reduced growth rate on YXT compared to the WT strain (**Figure 3A and C**). TLC assays indicated a noteworthy decrease in $AFB_1$ synthesis in the *xylPtorA* strain compared to the WT strain (**Figure 3B and F**). Additionally, the *xylPtorA* strain produced fewer conidia and sclerotium formation was absent on YXT medium (**Figure 3E**, **Figure 3—figure supplement 2A and B**). Interestingly, the *xylPtorA* strain exhibited a notable increase in sensitivity to rapamycin and SDS stress (**Figure 3G and H**). Collectively, these findings suggested that the TorA kinase plays a crucial role in various biological processes in *A. flavus*, including vegetative growth, asexual development, sclerotia formation, response to environmental stress, and biosynthesis of aflatoxins.

## The Sch9 kinase is involved in aflatoxin biosynthesis and the HOG pathway

Sch9, a Ser/Thr kinase belonging to the AGC family, is directly phosphorylated by TORC1. The phosphorylation of Sch9 is inhibited by rapamycin, as well as by carbon or nitrogen starvation (**Urban et al., 2007**). The putative *sch9* gene (AFLA_127440) in *A. flavus* encodes a protein consisting of 706 amino acids, exhibiting 80.31% identity to the Sch9 protein in *A. nidulans*. To investigate the role of Sch9 in *A. flavus*, we attempted to create a *sch9* deletion mutant using a homologous recombination strategy. The Δ*sch9* strain exhibited typical vegetative growth and conidiation similar to the WT strain (**Figure 4B**). TLC assay and quantitative analysis showed a significantly decreased aflatoxin production

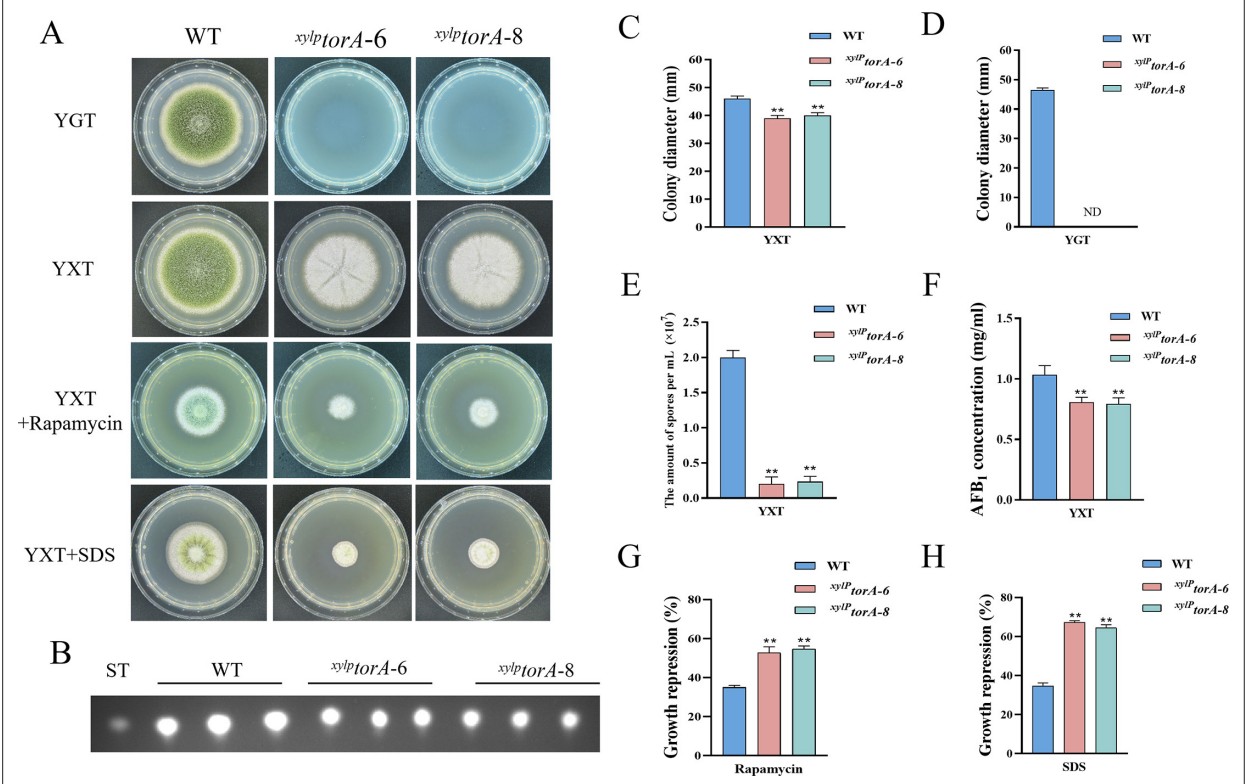

**Figure 3.** The TorA kinase plays a critical role in *A. flavus*. (**A**) Phenotype of the wild-type (WT) and the *xylptorA* strains in yeast extract-glucose agar (YGT) and yeast extract-xylose agar (YXT) medium amended with 100 ng/mL rapamycin and 300 mg/mL SDS at 37°C for 5 days. (**B**) Thin layer chromatography (TLC) assay of aflatoxin $B_1$ ($AFB_1$) production from the WT and the *xylptorA* strains cultured in YXT medium containing 1 g/L $MgSO_4 \cdot 7H_2O$ at 29°C for 6 days. (**C**) Statistical analysis of the colony diameter by the WT and the *xylptorA* strains in YXT medium. (**D**) Statistical analysis of the colony diameter by the WT and the *xylptorA* strains in YGT medium. (**E**) Conidial quantification of the WT and the *xylptorA* strains in YXT medium. (**F**) Quantitative analysis of $AFB_1$ as shown in (**B**). (**G**) The growth inhibition rate of the WT and the *xylptorA* strains under rapamycin. (**H**) The growth inhibition rate of the WT and the *xylptorA* strains under SDS stress. ND indicates no detection. * indicates that the significance level was p≤0.05, ** indicates that the significance level was p≤0.01, based on one-way ANOVA test with three replicates(n=3). Error bars represent the standard error of the mean (SEM).

The online version of this article includes the following figure supplement(s) for figure 3:

**Figure supplement 1.** Transcriptional levels of *torA*.

**Figure supplement 2.** TorA regulate sclerotia biosynthesis in *A. flavus*.

in the Δ*sch9* strain compared to the WT strain (*Figure 4C and G*). These results indicated that Sch9 regulates aflatoxin biosynthesis in *A. flavus*.

In yeast, Sch9 plays an important role in the transcriptional activation of osmostress-inducible genes (*Pascual-Ahuir and Proft, 2007*). To characterize the roles of Sch9 in stress response, we examined the sensitivity of the *sch9* deletion strain to various stresses including osmotic stress and calcium stress. Compared with the WT strain, the Δ*sch9* mutant significantly increased sensitivity to calcium stress (*Figure 4—figure supplement 1A and B*), and decreased sensitivity to osmotic stress induced by NaCl and KCl (*Figure 4D and H*). In *A. flavus*, the Sch9 kinase consists of several domains, including the protein kinase C conserved region 2 (C2), protein kinase (S_TKc), AGC-kinase C-terminal (S_TK_X) domain, and an ATP-binding site (*Figure 4A*). To explore the role of these domains and the ATP-binding site in Sch9 kinase, we constructed deletion strains for both domains and a *sch9*[K340A] mutant. The results revealed that the *sch9*[ΔS_TKc] strain, *sch9*[ΔS_TK_X] strain, and *sch9*[K340A] mutant exhibited similar phenotypes in osmotic stress and $AFB_1$ production as the Δ*sch9* mutant (*Figure 4B–D*). Therefore, we speculated that the protein kinase domain, kinase C-terminal domain, and the ATP-binding site at K340 play a crucial role in modulating the impact of Sch9 on aflatoxin biosynthesis and stress response in *A. flavus*.

To investigate the potential function of Sch9 in the HOG and TOR pathways in *A. flavus*, we conducted a sensitivity analysis of the Δ*sch9* strain to rapamycin. Compared to the WT strain, the

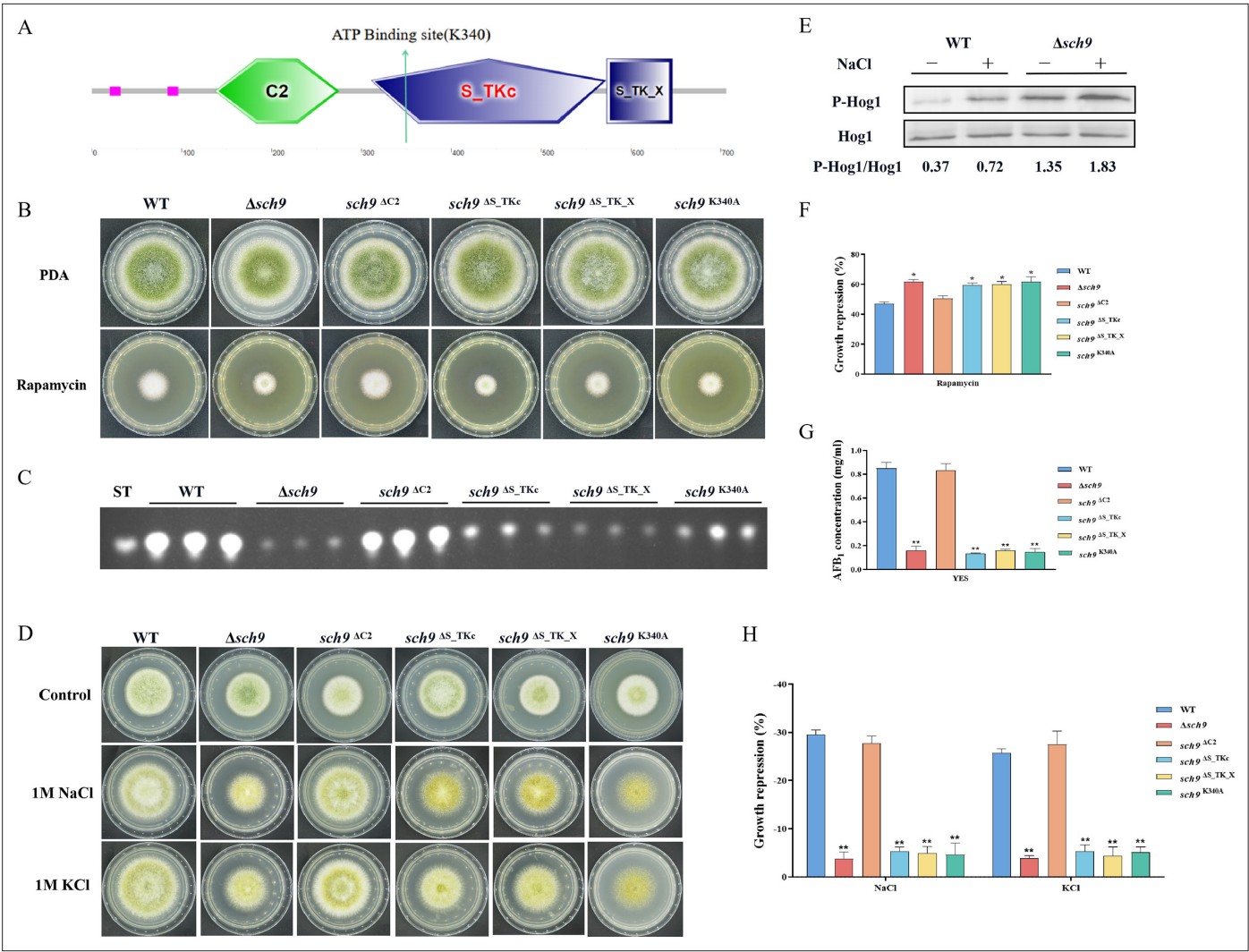

**Figure 4.** The Sch9 kinase participates in aflatoxin biosynthesis and the high osmolarity glycerol (HOG) pathway. (**A**) The structure diagram of the Sch9 kinase. (**B**) Phenotype of the wild-type (WT), $\Delta sch9$, $sch9^{\Delta C2}$, $sch9^{\Delta S\_TKc}$, $sch9^{\Delta S\_TK\_X}$, and $sch9^{K340A}$ strains grown on potato dextrose agar (PDA) medium amended with 100 ng/mL rapamycin at 37°C for 5 days. (**C**) Thin layer chromatography (TLC) assay of aflatoxin $B_1$ ($AFB_1$) production from the WT, $\Delta sch9$, $sch9^{\Delta C2}$, $sch9^{\Delta S\_TKc}$, $sch9^{\Delta S\_TK\_X}$, and $sch9^{K340A}$ strains cultured in yeast extract-sucrose agar (YES) liquid medium at 29°C for 6 days. (**D**) The phenotypes of the WT, $\Delta sch9$, $sch9^{\Delta C2}$, $sch9^{\Delta S\_TKc}$, $sch9^{\Delta S\_TK\_X}$, and $sch9^{K340A}$ strains on yeast extract-glucose agar (YGT) media amended with 1 M NaCl and 1 M KCl for 3 days. (**E**) The phosphorylation levels of Hog1 in the WT and $\Delta sch9$ strains were determined with or without osmotic stress. (**F**) The growth inhibition rate of the WT, $\Delta sch9$, $sch9^{\Delta C2}$, $sch9^{\Delta S\_TKc}$, $sch9^{\Delta S\_TK\_X}$, and $sch9^{K340A}$ strains under rapamycin stress. (**G**) Quantitative analysis of $AFB_1$ as shown in (**C**). (**H**) The growth inhibition rate of the WT, $\Delta sch9$, $sch9^{\Delta C2}$, $sch9^{\Delta S\_TKc}$, $sch9^{\Delta S\_TK\_X}$, and $sch9^{K340A}$ strains under osmotic stress. * indicates that the significance level was p≤0.05, ** indicates that the significance level was p≤0.01, based on one-way ANOVA test with three replicates(n=3). Error bars represent the standard error of the mean (SEM).

The online version of this article includes the following source data and figure supplement(s) for figure 4:

**Source data 1.** Original file for the western blot analysis in *Figure 4E* (anti-P-Hog1).

**Source data 2.** Original file for the western blot analysis in *Figure 4E* (anti-Hog1).

**Source data 3.** PDF containing *Figure 4E* and original scans of the relevant western blot analysis (anti-P-Hog1 and anti-Hog1) with highlighted bands and sample labels.

**Figure supplement 1.** Sch9 is involved in calcium stress.

*sch9* deletion strain exhibited a slightly higher sensitivity to rapamycin (*Figure 4B and F*). The MAKP Hog1 is an element of the HOG pathway. To verify the connection between Sch9 and the HOG-MAPK pathway, the phosphorylation levels of Hog1 kinase were measured under osmotic stress (1 M NaCl). Western blot analysis showed that the phosphorylation level of Hog1 increased significantly

in the Δ*sch9* strain (*Figure 4E*). This finding suggested that Sch9 plays a pivotal role in modulating the response of Hog1 to osmotic stress. Taken together, these results reflected that Sch9 regulates osmotic stress response via the HOG pathway in *A. flavus*.

## TapA and TipA participate in the regulation of cell wall stress response

Tap42, a protein that associates with phosphatase 2A, serves as the direct target of the Tor kinase in yeast. Tap42 plays a crucial role in various Tor functions, particularly in the regulation of transcriptional processes (*Düvel and Broach, 2004*). The genome of *A. flavus* contains a homolog of *tap42* named *tapA* (AFLA_092770), which is predicted to encode a 355 amino acid protein (sharing 31.9% identity with *S. cerevisiae* Tap42). To further determine the function of the TapA regulator in the TOR signaling pathway of *A. flavus*, we used homologous recombination to replace the original promoter with the *gpdA* promoter, a well-characterized strong promoter, to construct an overexpression mutant

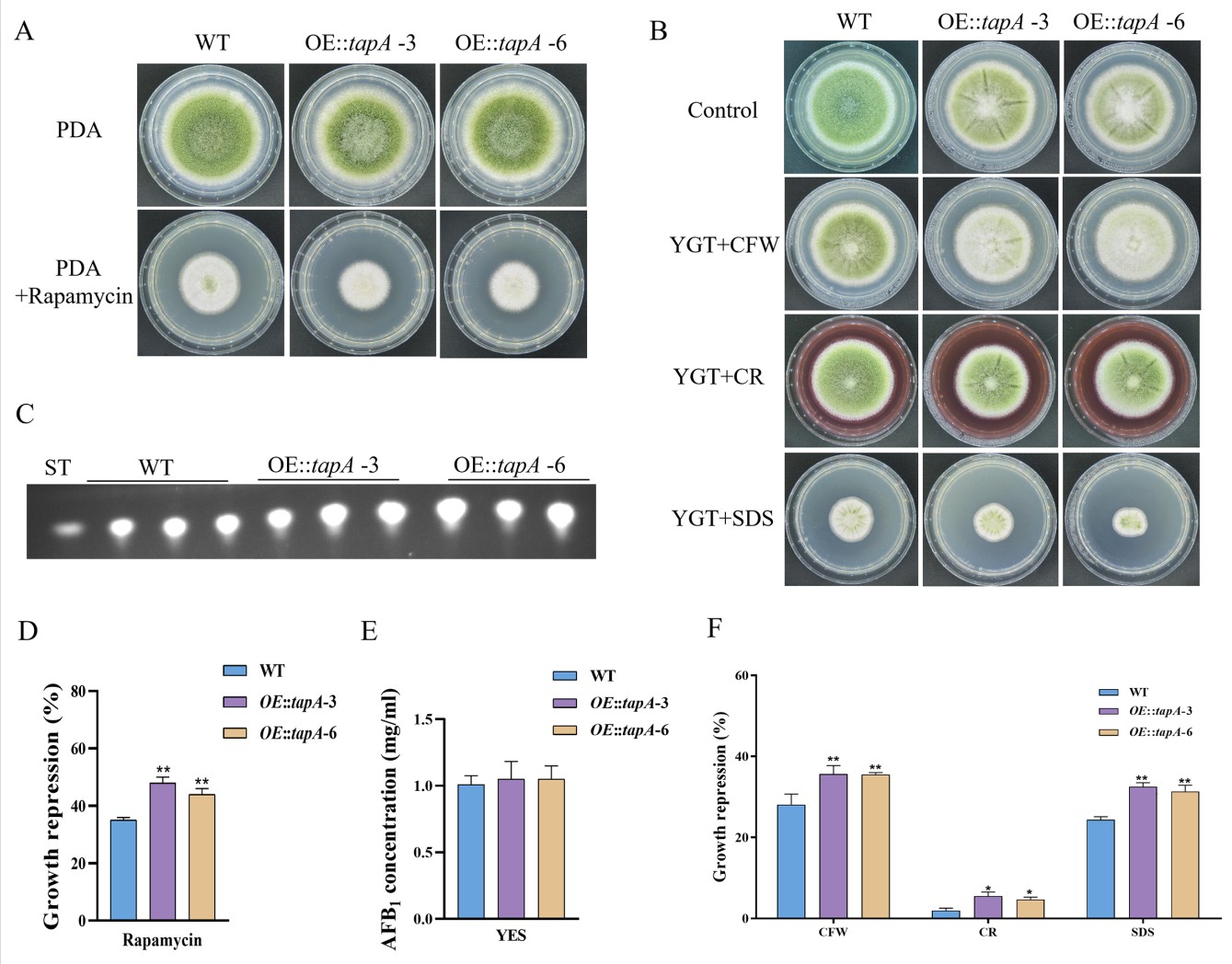

**Figure 5.** TapA regulates cell wall stress in *A. flavus*. (**A**) Phenotype of the wild-type (WT) and *OE::tapA* strains grown on potato dextrose agar (PDA) amended with 100 ng/mL rapamycin at 37°C for 5 days. (**B**) Colony morphology of the WT and *OE::tapA* strains grown on PDA media supplemented with 200 µg/mL Congo red (CR), 200 µg/mL Calcofluor white (CFW), or 300 µg/mL SDS at 37°C for 5 days. (**C**) Thin layer chromatography (TLC) assay of aflatoxin $B_1$ (AFB$_1$) production by the WT and *OE::tapA* strains cultured in yeast extract-sucrose agar (YES) liquid medium at 29°C for 6 days. (**D**) The growth inhibition rate of the WT and *OE::tapA* strains under rapamycin. (**E**) Quantitative analysis of AFB$_1$ as shown in (**C**). (**F**) The growth inhibition rate of the WT and *OE::tapA* strains under cell wall and cell membrane stress. * indicates that the significance level was p≤0.05, ** indicates that the significance level was p≤0.01, based on one-way ANOVA test with three replicates(n=3). Error bars represent the standard error of the mean (SEM).

The online version of this article includes the following figure supplement(s) for figure 5:

**Figure supplement 1.** Transcriptional levels of *tapA*.

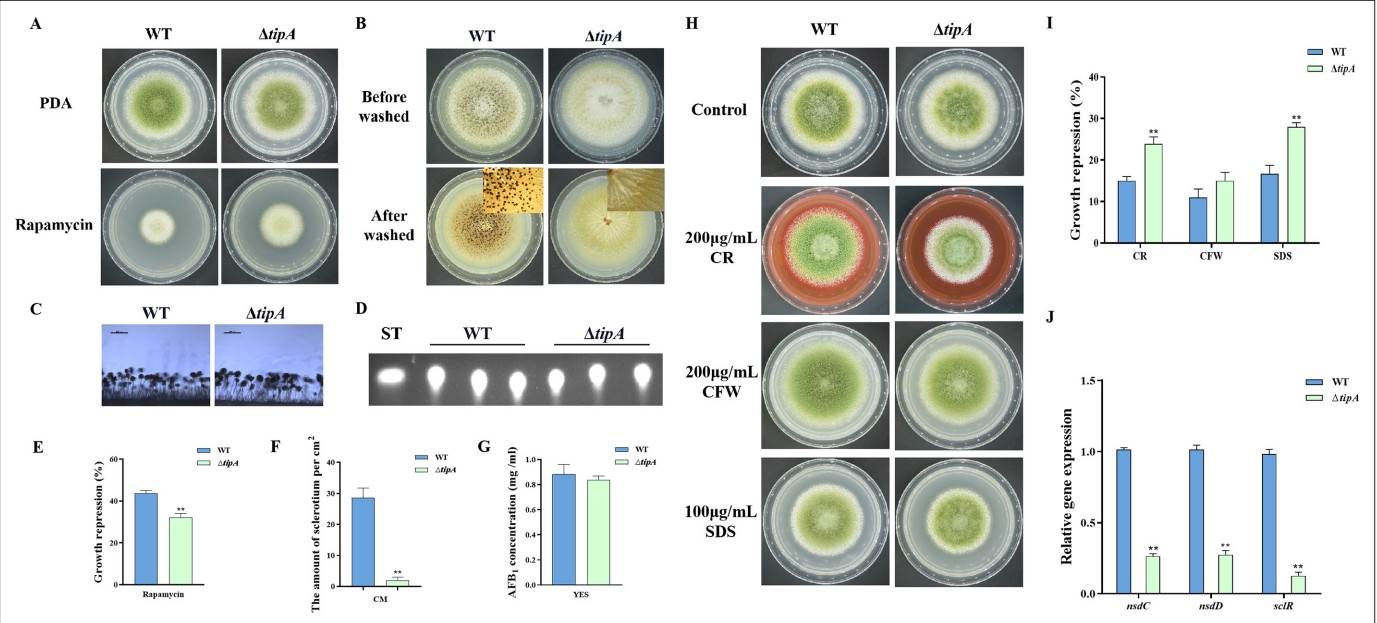

**Figure 6.** TipA regulates sclerotia development and cell wall stress in *A. flavus*. (**A**) Phenotype of the wild-type (WT) and Δ*tipA* strains grown on potato dextrose agar (PDA) amended with 100 ng/mL rapamycin at 37°C for 5 days. (**B**) Phenotypic characterization of the WT and Δ*tipA* strains grown on CM medium at 37°C for 7 days. (**C**) Morphology of conidiophores in the WT and Δ*tipA* strains observed by microscope. (**D**) Thin layer chromatography (TLC) assay of aflatoxin B$_1$ (AFB$_1$) production by the WT and Δ*tipA* strains cultured in yeast extract-sucrose agar (YES) liquid medium at 29°C for 6 days. (**E**) The growth inhibition rate of the WT and Δ*tipA* strains under rapamycin stress. (**F**) Amount of sclerotia produced by the WT and Δ*tipA* strains. (**G**) AFB$_1$ quantitative analysis of the WT and Δ*tipA* strains described in (**D**). (**H**) Colony morphology of the WT and Δ*tipA* strains grown on PDA media supplemented with 200 μg/mL Congo red (CR), 200 μg/mL Calcofluor white (CFW), and 100 μg/mL SDS at 37°C for 5 days. (**I**) The growth inhibition rate of the WT and Δ*tipA* strains under cell wall and cell membrane stress. (**J**) Relative expression levels of sclerotia formation genes in the WT and Δ*tipA* strains. ** indicates that the significance level was p≤0.01, based on one-way ANOVA test with three replicates(n=3). Error bars represent the standard error of the mean (SEM).

(*OE::tapA*). The transformants of the mutant were verified using PCR. Quantitative expression analysis of the *tapA* gene in the overexpressed strains revealed a substantial upregulation compared to the WT strain (*Figure 5—figure supplement 1*). Our results showed that the colony diameter of the *OE::tapA* mutant was not significantly changed compared to the WT strain (*Figure 5A*). However, the rapamycin sensitivity assay indicated that the *OE::tapA* strain displayed increased sensitivity to rapamycin (*Figure 5A and D*). We found no significant difference in aflatoxin production between the *OE::tapA* and the WT strains (*Figure 5C and E*), suggesting that TapA may not play a direct role in the regulation of aflatoxin biosynthesis. In addition, the *OE::tapA* strain was more sensitive to cell wall and cell membrane stress, including SDS, CFW, and Congo red (CR) (*Figure 5B and F*). These findings have demonstrated the significantly positive impact of TapA on cell wall stress in *A. flavus*.

In *S. cerevisiae*, Tip41 was identified as a Tap42-interacting protein and has been measured to function as a negative regulator of Tap42, which downregulates TORC1 signaling via activation of PP2A phosphatase (*Jacinto et al., 2001*). The putative ortholog of *tip41* in *A. flavus* is named *tipA* (AFLA_047310). The amino acid sequence of TipA exhibits a similarity of 40.49% to Tip41 in *S. cerevisiae*. To elucidate the function of the TOR signaling pathway protein TipA in *A. flavus*, we generated the *tipA* deletion mutants. The Δ*tipA* strain exhibited a slightly reduced number of conidia produced on PDA compared to the WT strain (*Figure 6A and C*). Meanwhile, the Δ*tipA* strain exhibited decreased sensitivity to rapamycin (*Figure 6A and E*). The Δ*tipA* strain was unable to produce any sclerotia (*Figure 6B and F*). Whereas Δ*tipA* strain do not affect aflatoxin production (*Figure 6D and G*). In addition, the Δ*tipA* strain was more sensitive to the cell wall and cell membrane stress, including CR and SDS (*Figure 6H and I*). Additionally, the expression levels of *nsdC*, *nsdD*, and *sclR* exhibited a decrease in Δ*tipA* strain compared to the WT strain (*Figure 6J*). These results indicated that TipA is crucial for the sclerotial formation and cell wall stress in *A. flavus*.

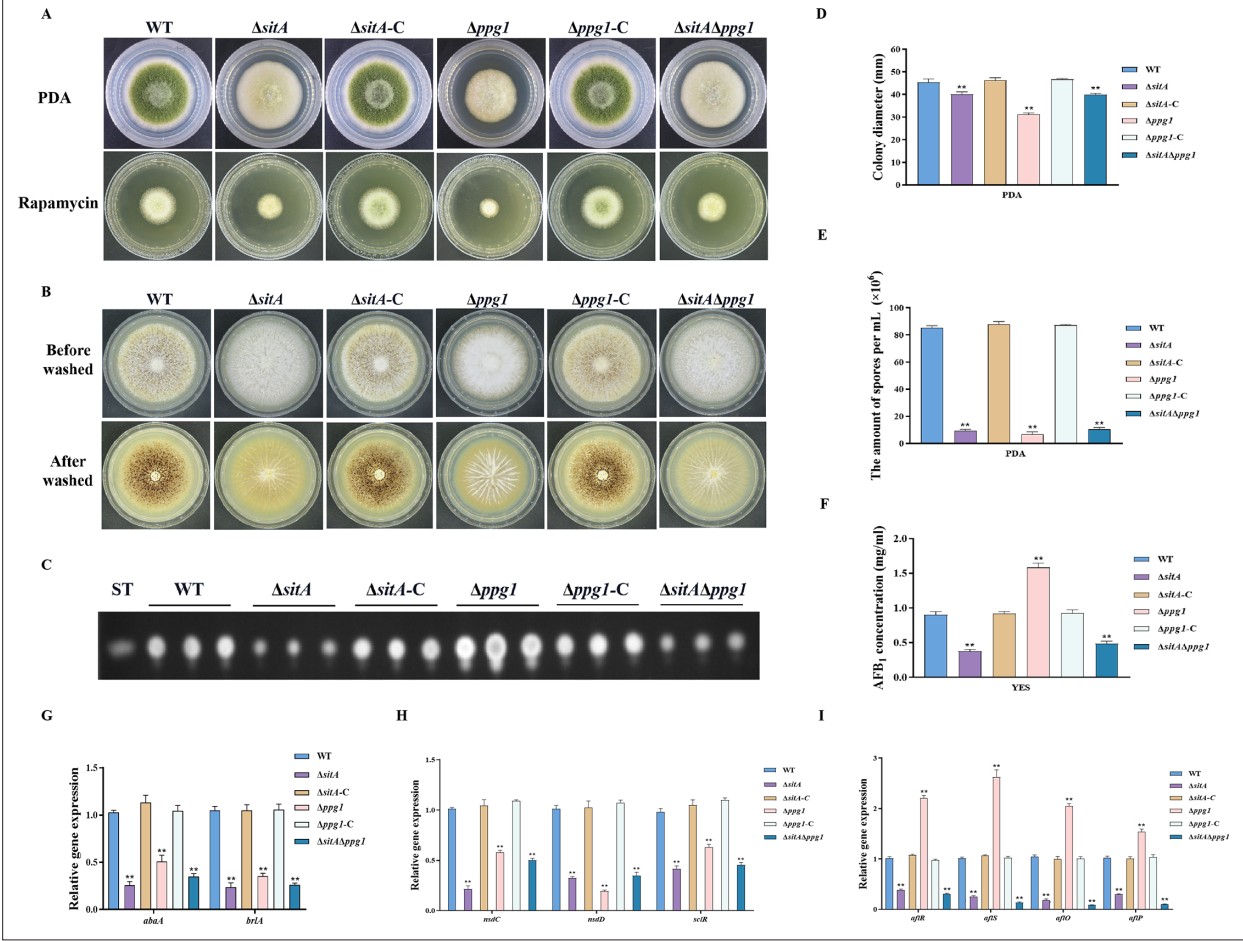

**Figure 7.** The impact of the phosphatases SitA and Ppg1 on the growth, conidiation, and aflatoxin biosynthesis in *A. flavus*. (**A**) Colony morphology of the wild-type (WT), single knockout, and double knockout strains cultured on potato dextrose agar (PDA) medium amended with 100 ng/mL rapamycin at 37°C for 5 days. (**B**) Colony morphology of the WT, single knockout, and double knockout strains cultured on CM medium at 37°C for 7 days. (**C**) Thin layer chromatography (TLC) analysis of aflatoxin B₁ (AFB₁) production from the WT, single knockout, and double knockout strains cultured in yeast extract-sucrose agar (YES) liquid medium at 29°C for 6 days. (**D**) Growth diameter of the WT, single knockout, and double knockout strains on PDA media. (**E**) Conidial quantification of the WT, single knockout, and double knockout strains. (**F**) AFB₁ quantitative analysis of the WT, single knockout, and double knockout strains. (**G**) Relative expression levels of conidia synthesis genes in the WT, single knockout, and double knockout strains. (**H**) Relative expression levels of sclerotia synthesis genes in the WT, single knockout, and double knockout strains. (**I**) Relative expression levels of aflatoxin biosynthesis genes in the WT, single knockout, and double knockout strains. ** indicates that the significance level was $p \leq 0.01$, based on one-way ANOVA test with three replicates(n=3). Error bars represent the standard error of the mean (SEM).

## Phosphatase SitA and Ppg1 are involved in growth, conidiation, and sclerotial formation

In *S. cerevisiae*, Tap42 has been identified to interact with the catalytic subunit of type 2A protein phosphatases, including Pph3, Pph21, and Pph22, as well as the type 2A-like phosphatases Sit4 and Ppg1 (*Wang et al., 2003*). Phosphatase *sitA* (AFLA_108450) and *ppg1* (AFLA_132610) were identified in *A. flavus*, encoding putative orthologs of *sit4* and *ppg1* in *S. cerevisiae*. To determine the biological functions of phosphatases SitA and Ppg1 in *A. flavus*, we disrupted the *sitA* and *ppg1* by homologous recombination strategy. To further investigate the relationship between phosphatases SitA and Ppg1, we additionally generated complemented and double deletion mutants. The vegetative development phenotype analysis showed that the Δ*sitA*, Δ*ppg1*, and Δ*sitA*/Δ*ppg1* strains grew significantly more slowly than the WT strain on PDA (*Figure 7A and D*). Compared with the WT strain, the Δ*sitA*, Δ*ppg1*, and Δ*sitA*/Δ*ppg1* strains were more sensitive to rapamycin (*Figure 7A*). In addition, the Δ*sitA*, Δ*ppg1*, and Δ*sitA*/Δ*ppg1* strains formed lower number of conidiophores (*Figure 7A*). To gain further insight into the role of SitA and Ppg1 in the process of conidiation, qRT-PCR was employed to determine the

transcript levels of conidiation-related genes (*abaA* and *brlA*). The expression levels of the *abaA* and *brlA* genes were significantly downregulated in the Δ*sitA*, Δ*ppg1*, and Δ*sitA*/Δ*ppg1* mutants when compared to the WT and complemented strains (*Figure 7G*). The analysis of sclerotial formation revealed that the Δ*sitA*, Δ*ppg1*, and Δ*sitA*/Δ*ppg1* mutants exhibited a complete inability to produce any sclerotia (*Figure 7B*). To further prove these findings, we conducted qRT-PCR to determine the expression levels of three regulators (*nsdC*, *nsdD*, and *sclR*) of sclerotia development. Our findings indicated that the expression levels of *nsdC*, *nsdD*, and *sclR* were all significantly reduced in the Δ*sitA*, Δ*ppg1*, and Δ*sitA*/Δ*ppg1* mutants (*Figure 7H*), which is consistent with the observed decrease in sclerotia production in the Δ*sitA*, Δ*ppg1*, and Δ*sitA*/Δ*ppg1* mutants. These results showed that SitA and Ppg1 are crucial for hyphal development, conidiation, and sclerotia formation in *A. flavus*.

## SitA and Ppg1 play crucial roles in aflatoxin biosynthesis and pathogenicity

We assayed AFB$_1$ biosynthesis in the Δ*sitA*, Δ*ppg1*, and Δ*sitA*/Δ*ppg1* mutants. TLC analyses revealed that the Δ*sitA* and Δ*sitA*/Δ*ppg1* strains exhibited a decrease in AFB$_1$ production, whereas the Δ*ppg1* strain showed a significant increase in AFB$_1$ production compared to the WT and complemented strains (*Figure 7C and F*). The expression levels of genes related to aflatoxin synthesis, including structural genes (*aflO* and *aflP*) and regulatory genes (*aflR* and *aflS*), were determined by qRT-PCR in the Δ*sitA*, Δ*ppg1*, and Δ*sitA*/Δ*ppg1* mutants. The results revealed a significant downregulation of these genes in the Δ*sitA* and Δ*sitA*/Δ*ppg1* mutants, while upregulated in the Δ*ppg1* mutant, compared to the WT and complemented strains (*Figure 7I*). These results indicated that SitA and Ppg1 play different roles in regulating aflatoxin biosynthesis in *A. flavus*.

We conducted pathogenicity tests on peanuts and maize to examine the pathogenicity of the phosphatases SitA and Ppg1 on crops. The results showed that the WT and complemented strains demonstrated complete virulence on all peanut and maize seeds. In contrast, the Δ*sitA*, Δ*ppg1*, and Δ*sitA*/Δ*ppg1* mutants displayed compromised colonization on peanut and maize seeds (*Figure 8A*). In addition, the number of conidial productions in these mutants on the infected peanut seeds also dramatically decreased compared to those of the WT and complemented strains (*Figure 8C*). The TLC analyses revealed that the Δ*ppg1* mutant exhibited increased production of AFB$_1$ in peanuts and maize. Conversely, the levels of aflatoxin produced by the Δ*sitA* and Δ*sitA*/Δ*ppg1* mutants on peanut and maize were significantly reduced compared to the WT and complemented strains (*Figure 8B and D*). This observation aligns with the above results of aflatoxin biosynthesis from all mutant strains in the yeast extract-sucrose agar (YES) medium. All these data showed that SitA and Ppg1 are crucial for crop seeds' pathogenicity.

## SitA and Ppg1 are involved in regulating CWI

To explore the role of phosphatases SitA and Ppg1 in response to cell wall stress, all strains were cultivated on PDA amended with CFW, CR, and SDS. We found that the relative growth inhibition of Δ*sitA*, Δ*ppg1*, and Δ*sitA*/Δ*ppg1* mutants induced by cell wall stress was significantly increased compared to the WT and complemented strains (*Figure 9A and B*). Further investigations using qRT-PCR revealed a notable reduction in the expression levels of genes related to the synthesis of cell walls, specifically chitin synthase genes *chsA*, *chsD*, and *chsE* in the Δ*sitA*, Δ*ppg1*, and Δ*sitA*/Δ*ppg1* strains (*Figure 9C*). Western blot analysis also revealed that the phosphorylation level of Slt2 was significantly elevated in the Δ*sitA*, Δ*ppg1*, and Δ*sitA*/Δ*ppg1* strains under cell wall stress (200 µg/mL CFW) (*Figure 9D*). In conclusion, the phosphatases SitA and Ppg1 play a crucial role in regulating CWI.

## The phosphatase complex Nem1/Spo7 is important for growth, aflatoxin biosynthesis, and LD biogenesis

LDs are highly dynamic spherical organelles, which appear in the cytosol of most eukaryotic cell types (*Sánchez-Álvarez et al., 2022*). Rapamycin treatment led to the accumulation of LDs in various filamentous fungi (*Liu et al., 2019*). The FgTOR signaling cascade is implicated in the regulation of FgNem1 phosphorylation profile, predominantly mediated by the sequential activation of the FgPpg1/Sit4 and FgCak1 kinase modules, along with additional kinases in *F. graminearum* (*Liu et al., 2019*). In this study, we observed that the treatment of hyphae with rapamycin resulted in the accumulation of LD biogenesis (*Figure 10A*). This finding suggested that the TOR pathway may regulate

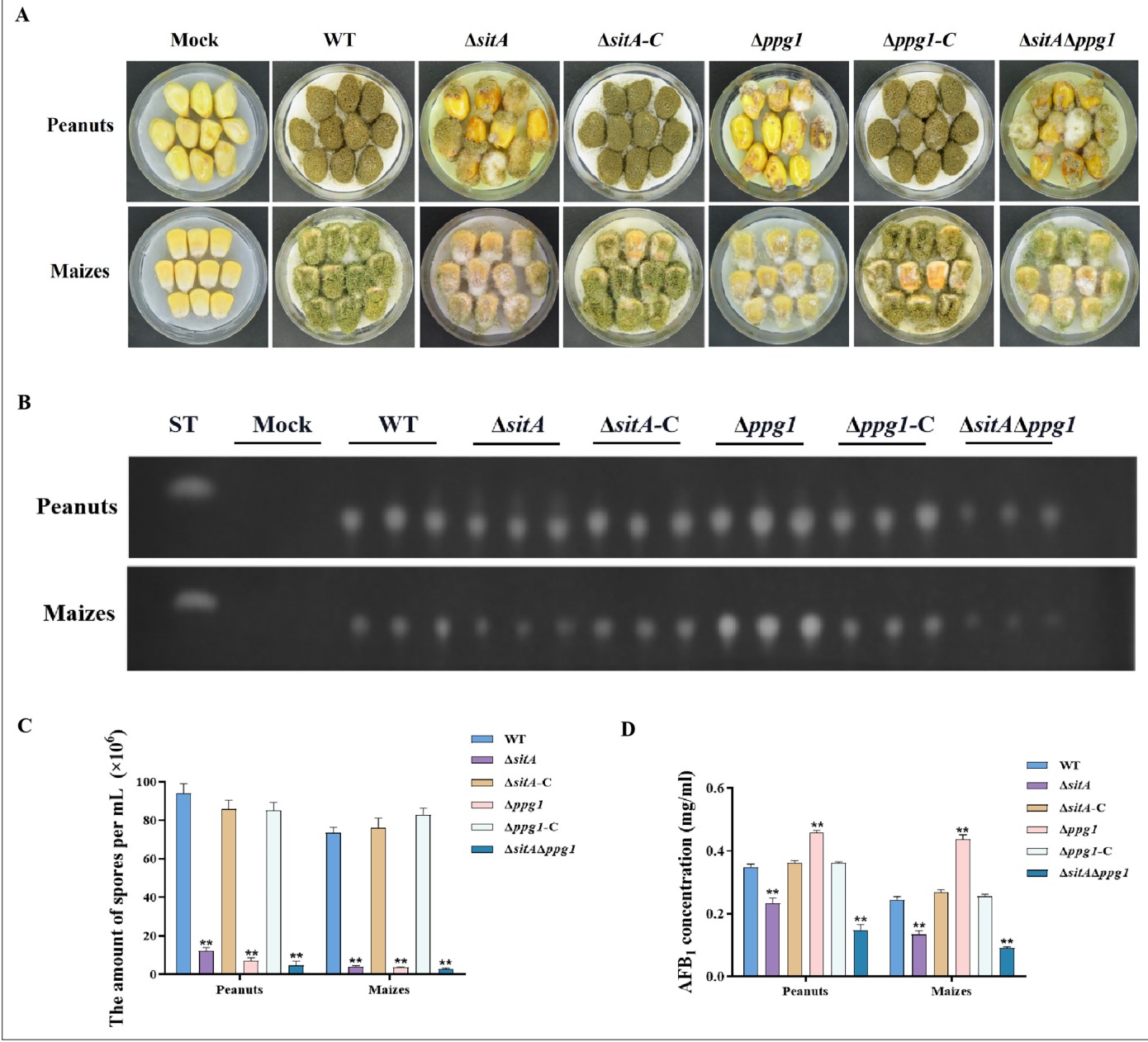

**Figure 8.** Pathogenicity analysis of the phosphatases SitA and Ppg1 in *A. flavus*. (**A**) Phenotypes of peanut and maize infected by the wild-type (WT), single knockout, and double knockout strains. (**B**) Thin layer chromatography (TLC) was used to detect the aflatoxin B$_1$ (AFB$_1$) from peanut and maize infected by the WT, single knockout, and double knockout strains. (**C**) Quantitative analysis of conidia production from peanut and maize infected by the WT, single knockout, and double knockout strains. (**D**) Quantitative analysis of AFB$_1$ from peanut and maize infected by the WT, single knockout, and double knockout strains. ** indicates that the significance level was p≤0.01, based on one-way ANOVA test with three replicates(n=3). Error bars represent the standard error of the mean (SEM).

the LD biogenesis. However, the underlying regulatory mechanism of LD biosynthesis remains largely unknown in *A. flavus*. To investigate the regulatory mechanism of the TOR pathway in *A. flavus* that controls LD biogenesis, we examined LD accumulation in Δ*sch9*, Δ*sitA*, and Δ*ppg1* mutants. BODIPY staining assays revealed that LD biogenesis was not observed in the Δ*sitA* and Δ*ppg1* strains upon treatment with rapamycin, in contrast to the WT and Δ*sch9* strains (*Figure 10A–D*). These results indicated that LD biogenesis induced by rapamycin is predominantly reliant on the phosphatases SitA and Ppg1 in *A. flavus*. In *F. graminearum*, the phosphatase FgNem1/FgSpo7 complex was involved in the induction of LD biogenesis by rapamycin (*Liu et al., 2019*). Phosphatase *nem1* (AFLA_002649) and

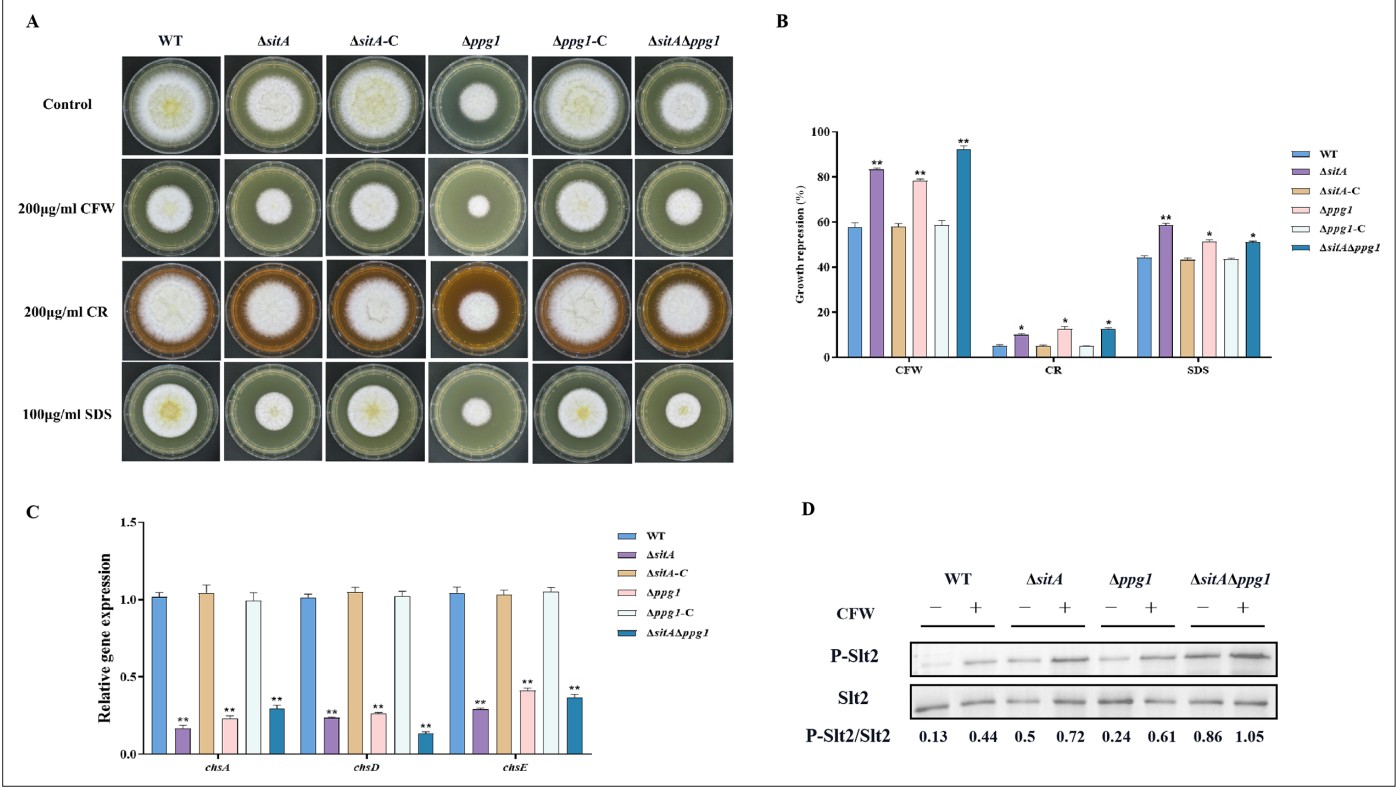

**Figure 9.** Sensitivity of phosphatases SitA and Ppg1 to cell wall damaging agents in *A. flavus*. (**A**) Morphology of the wild-type (WT), single knockout, and double knockout strains grown on yeast extract-sucrose agar (YES) media supplemented with 200 µg/mL Congo red (CR), 200 µg/mL Calcofluor white (CFW), or 100 µg/mL SDS at 37°C for 5 days. (**B**) The growth inhibition rate of the WT, single knockout, and double knockout strains under cell wall and cell membrane stress. (**C**) Relative expression levels of cell wall synthesis genes (*chsA*, *chsD*, and *chsE*) in the WT, single knockout, and double knockout strains. (**D**) The phosphorylation level of Slt2 in the WT, single knockout, and double knockout strains was detected with or without cell wall stress. * indicates that the significance level was p≤0.05, ** indicates that the significance level was p≤0.01, based on one-way ANOVA test with three replicates(n=3). Error bars represent the standard error of the mean (SEM).

The online version of this article includes the following source data for figure 9:

**Source data 1.** Original file for the western blot analysis in *Figure 9D* (anti-P-Slt2).

**Source data 2.** Original file for the western blot analysis in *Figure 9D* (anti-Slt2).

**Source data 3.** PDF containing *Figure 9D* and original scans of the relevant western blot analysis (anti-P-Slt2 and anti-Slt2) with highlighted bands and sample labels.

*spo7* (AFLA_028590) were identified in *A. flavus*, which encode putative orthologs of *nem1* and *spo7* in *F. graminearum*. To examine the role of the *nem1* and *spo7* genes in *A. flavus*, we generated Δ*nem1* and Δ*spo7* mutants using homologous recombination. The Δ*nem1* and Δ*spo7* mutants displayed a notable reduction in both vegetative growth and conidiation compared to the WT strain (*Figure 11A, C, and D*). TLC assay revealed a significantly decreased aflatoxin production in the Δ*nem1* and Δ*spo7* compared to the WT strain (*Figure 11B*). In addition, BODIPY staining showed no visible LDs in the hyphae of Δ*nem1* and Δ*spo7* after rapamycin treatment (*Figure 11E*). Taken together, these results indicated that Nem1/Spo7 are involved in vegetative growth, conidiation, aflatoxin production, and LD biogenesis.

## Discussion

Aflatoxins are a class of toxic secondary metabolites synthesized by *A. flavus*, including aflatoxin B$_1$, B$_2$, G$_1$, and G$_2$, among which AFB$_1$ has been regarded as the most pathogenic and toxic aflatoxin (*Gizachew et al., 2019*). Therefore, understanding the regulatory mechanisms involved in fungal secondary metabolites biosynthesis would be instrumental for effectively addressing aflatoxin control in *A. flavus*. In this study, we demonstrated that the TOR signaling pathway plays a pivotal role in

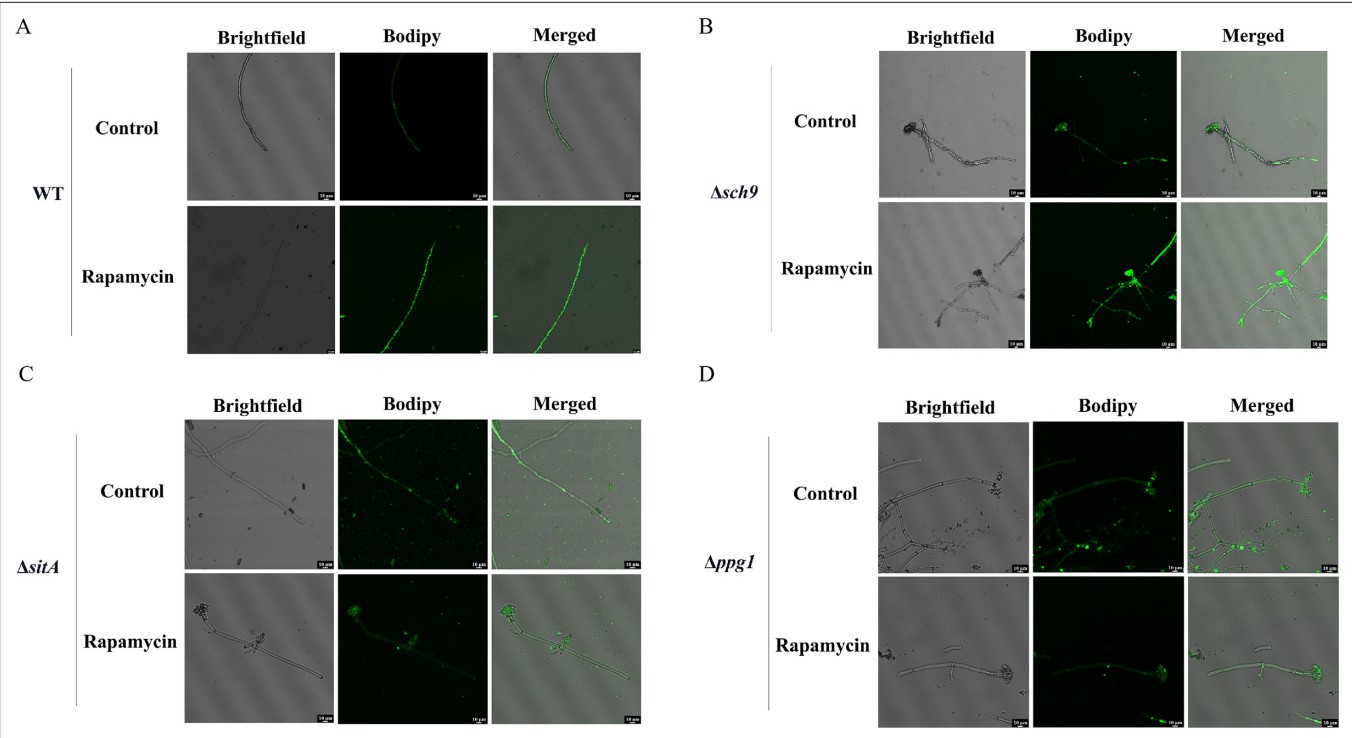

**Figure 10.** Phosphatases SitA and Ppg1 are involved in regulating lipid droplet biogenesis. (**A**) The intracellular lipid droplets were stained by boron dipyrromethene difluoride (BODIPY) in hyphae of the wild-type (WT) strain and observed under fluorescence microscopy. The phenotype of lipid droplets accumulation in the mycelia of the WT strain treated with or without 100 ng/mL rapamycin for 6 hr (Scale bar, 10 µm). (**B**) Phenotype of lipid droplets accumulation in the mycelia of the Δ*sch9* strain treated with or without 100 ng/mL rapamycin for 6 hr (Scale bar, 10 µm). (**C**) Phenotype of lipid droplets accumulation in the mycelia of the Δ*sitA* strain treated with or without 100 ng/mL rapamycin for 6 hr (Scale bar, 10 µm). (**D**) Phenotype of lipid droplets accumulation in the mycelia of the Δ*ppg1* strain treated with or without 100 ng/mL rapamycin for 6 hr (Scale bar, 10 µm).

regulating growth, sporulation, sclerotia formation, aflatoxin production, and various stress responses in *A. flavus*. Through conducting functional studies on genes within the TOR signaling pathway, we explored the conservation and complexity exhibited by the TOR signaling pathway in *A. flavus*. In addition, we constructed a crosstalk network between the TOR and other signaling pathways (HOG, CWI) and analyzed the specific regulatory process. We proposed that the TOR signaling pathway interacts with multiple signaling pathways, including HOG and CWI pathways, to modulate the cellular responses to diverse environmental stresses (*Figure 12*).

Our findings indicated that the TOR signaling pathway is pivotal in *A. flavus*, we have observed that rapamycin has a significant impact on the growth and sporulation of *A. flavus* (*Figure 1A*). Additionally, it exhibits a strong inhibitory effect on sclerotia production and aflatoxin biosynthesis (*Figure 1B and C*). The inhibitory effect has been documented in various fungal species, including *C. albicans* (*Cruz et al., 2001*), *Cryptococcus neoformans* (*Cruz et al., 2001*), *P. anserine* (*Dementhon et al., 2003*), *A. nidulans* (*Fitzgibbon et al., 2005*), *M. oryzae* (*Qian et al., 2018*), and *F. graminearum* (*Yu et al., 2014*). This suggested that the TOR signaling pathway may have a conserved function in filamentous fungi. The number of FK506-binding proteins varies among different species: there are four Fkbps proteins in *S. cerevisiae* and three Fkbps proteins in *S. pombe* (*Galat, 2013*). In *S. cerevisiae*, rapamycin can bind to Fkbp protein, resulting in the irreversible inhibition of the $G_1$ phase of the cell cycle and the regulation of cell growth (*Kasahara, 2021*). In *A. nidulans*, the *fkbp12* gene homolog *fprA* displays considerable sequence similarity with homologs identified in the *S. cerevisiae* (*Fitzgibbon et al., 2005*). In this study, based on the homologous alignment of the protein sequence of FprA, it was observed that four *fkbp* genes are present in *A. flavus*. Among these, deletion of *fkbp3* exhibited enhanced tolerance to rapamycin (*Figure 2A*). So we speculated that rapamycin forms a complex with Fkbp3 in *A. flavus*, subsequently targeting downstream target genes to exert its biological function. We found that Fkbp3 is involved in sclerotia formation and aflatoxin biosynthesis in *A. flavus* (*Figure 2A*, *Figure 2—figure supplement 2*), and subsequent analysis revealed that the

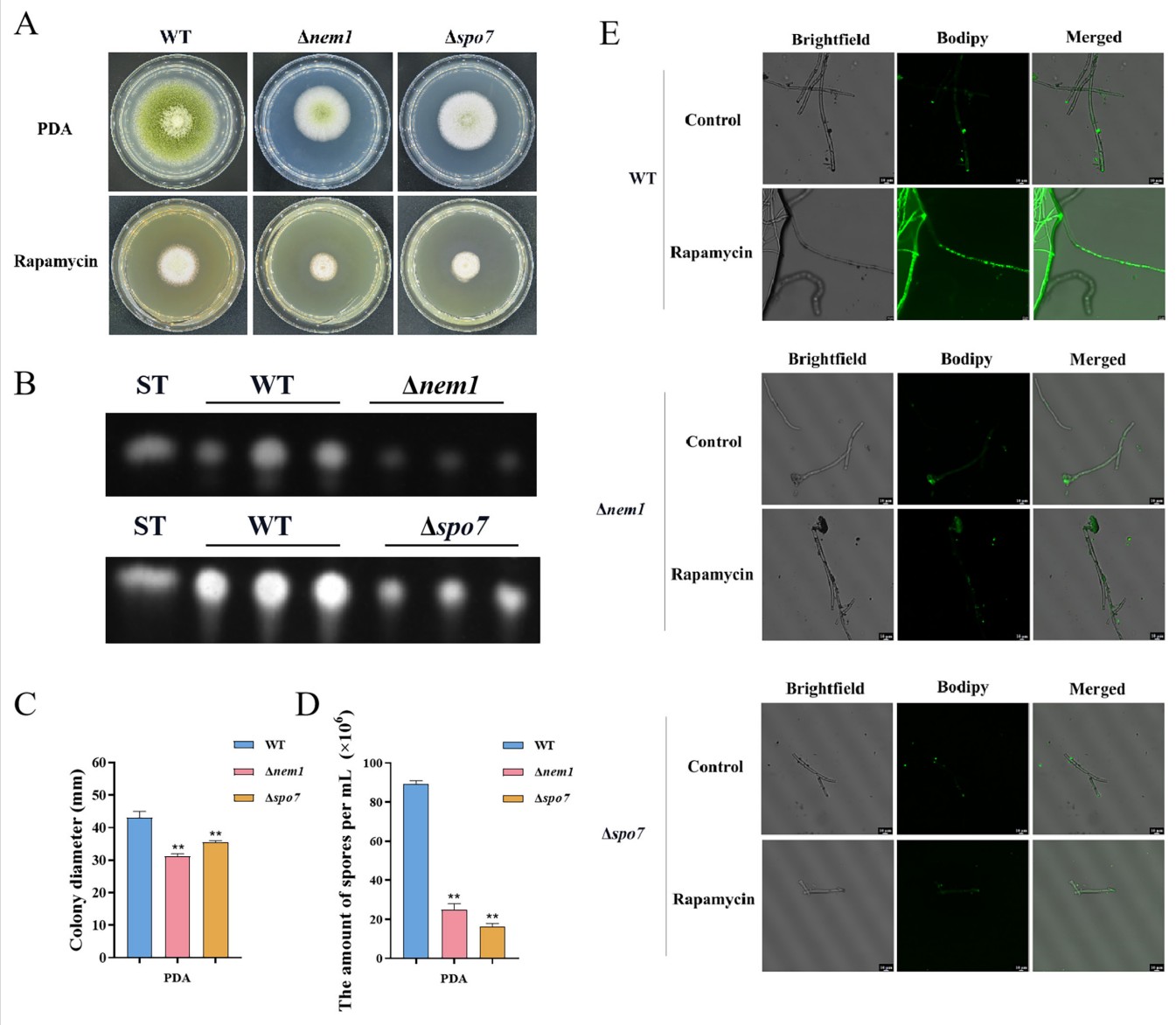

**Figure 11.** Phosphatase complex Nem1/Spo7 play significant role in growth, conidiation, aflatoxin, and lipid droplet biogenesis. (**A**) Phenotype of the wild-type (WT), Δ*nem1*, and Δ*spo7* strains grown on potato dextrose agar (PDA) amended with 100 ng/mL rapamycin at 37°C for 5 days. (**B**) Thin layer chromatography (TLC) assay of aflatoxin $B_1$ ($AFB_1$) production from the WT, Δ*nem1*, and Δ*spo7* strains cultured in yeast extract-sucrose agar (YES) liquid medium at 29°C for 6 days. (**C**) Growth diameter of the WT, Δ*nem1*, and Δ*spo7* strains on PDA media. (**D**) Statistical analysis of the sporulation by the WT, Δ*nem1*, and Δ*spo7* strains. (**E**) Phenotype of lipid droplets accumulation in the mycelia of the WT, Δ*nem1*, and Δ*spo7* strains treated with or without 100 ng/mL rapamycin for 6 hr (Scale bar, 10 μm). ** indicates that the significance level was p≤0.01, based on one-way ANOVA test with three replicates(n=3). Error bars represent the standard error of the mean (SEM).

underlying mechanism may be associated with succinylation modification. The succinylation sites of Fkbp3 were identified through the utilization of succinylation proteomics data. By constructing point mutant strains and conducting phenotype experiments, we found that the succinylation of Fkbp3 at residue K19 is involved in rapamycin resistance and aflatoxin biosynthesis (*Figure 2E and G*). Our results provided new insights to the understanding of post-translational modifications in the regulation of TOR pathway components. Previous studies have demonstrated that Fkbp12 has been identified as the receptor for FK506 and rapamycin, which inhibits calcineurin and TORC1, respectively (*Kasahara, 2021*). The deletion of Fkbp12 confers the resistance to rapamycin in *C. albicans* (*Cruz et al., 2001*) and *C. neoformans* (*Cruz et al., 2001*). The entomopathogenic fungus *Beauveria bassiana* contains three putative *fkbp* genes, and disruption of *Bbfkbp12* significantly increases the resistance

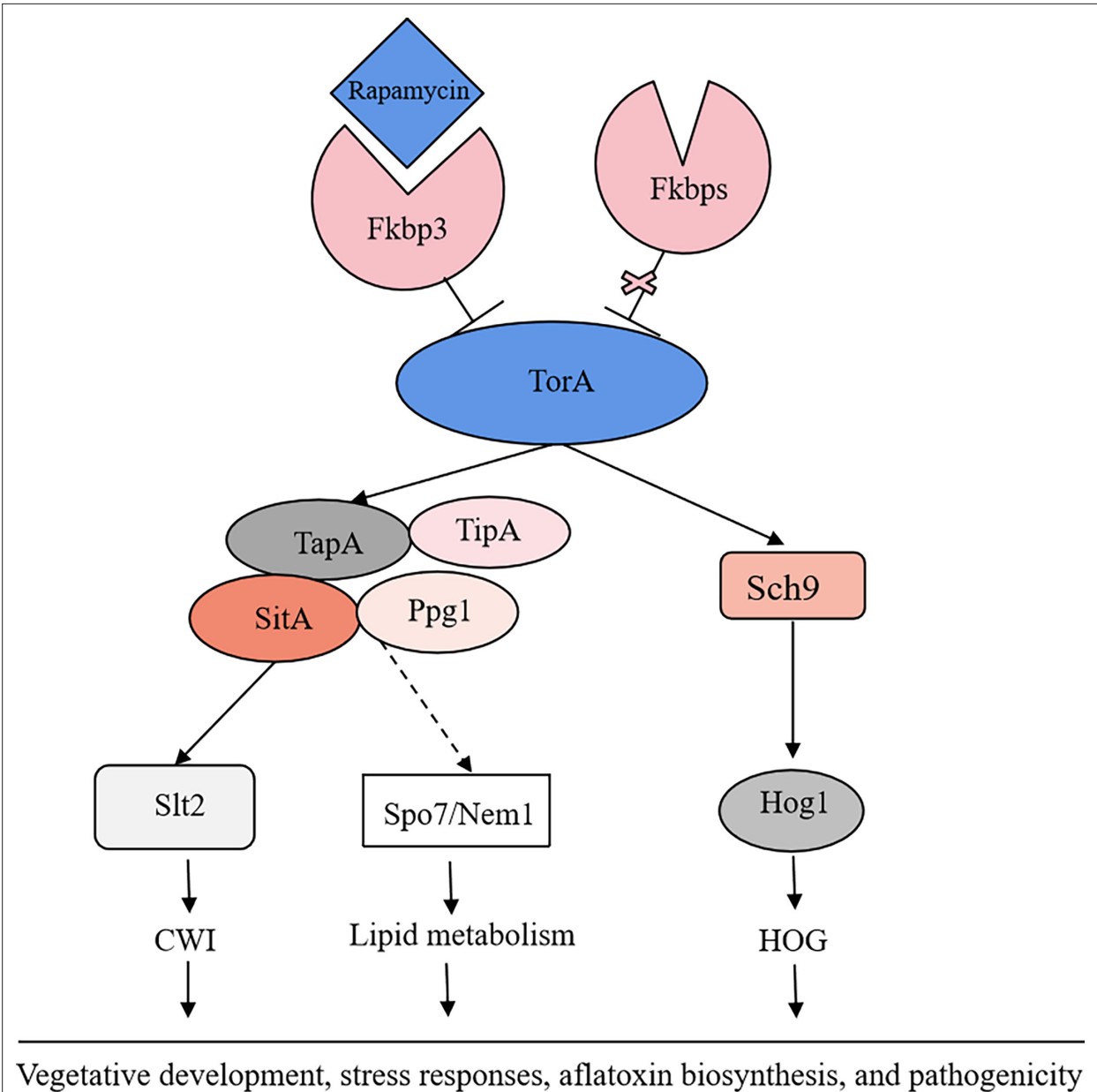

**Figure 12.** The proposed model of the target of rapamycin (TOR) pathway in *A. flavus*. Based on the above results, we propose a hypothesis regarding the TOR signaling pathway model in *A. flavus*. Rapamycin forms a complex with Fkbp3, and this particular complex can bind to the TorA kinase, thereby impeding its regular functionality. Sch9, functioning as a downstream element of the TorA kinase, regulates aflatoxin biosynthesis and high osmolarity glycerol (HOG) signaling pathway. As another important target of the TorA kinase, TapA-phosphatase complex is involved in the regulation of growth, sporulation, sclerotia, aflatoxin production, cell wall integrity (CWI) signaling pathway, lipid droplet synthesis, and other processes. In conclusion, the TOR signaling pathway plays a crucial role in various aspects of *A. flavus*, including vegetative development, stress response, aflatoxin biosynthesis, and pathogenicity.

to rapamycin and FK506 (*Li et al., 2020*). The plant pathogenic fungus *F. graminearum* harbors three Fkbps proteins, among which deletion of *Fgfkbp12* leads to resistance to rapamycin (*Yu et al., 2014*). Thus, we proposed that these Fkbps may exhibit analogous functionalities in conferring resistance to rapamycin. In *B. cinerea*, deletion of *Bcfkbp12* caused a reduction of the virulence of strain T4, while knockout of the *fkbp12* gene did not affect the pathogenic development of strain B05.10 (*Meléndez et al., 2009*). These findings demonstrated that the *fkbp12* homologous genes exhibited conservation in terms of rapamycin resistance. However, there is significant variation in other biological functions among fungal species, which may be attributed to species or strain evolutionary divergence. Our

study indicated the pivotal role of Fkbp3 in modulating the effects of rapamycin on *A. flavus*, Fkbp3 is also crucial for aflatoxin biosynthesis. Targeted inhibition of Fkbp3 could potentially suppress aflatoxin production, given its significance, Fkbp3 can be developed as a fungicide target to reduce aflatoxin biosynthesis, thereby reducing the risk of aflatoxin contamination in agricultural products.

The Tor kinase functions as a fundamental component of the TOR signaling pathway, which exhibits evolutionary conservation across fungi, plants, and mammals (*Xiong and Sheen, 2014*). In *S. cerevisiae*, the Tor kinase can interact with a diverse range of proteins, resulting in the formation of complex structures known as TORC1 and TORC2 (*Weisman, 2016*). TORC1 can detect various signals, including nutrients, growth factors, and environmental stress. It plays a crucial role in regulating cellular processes such as gene transcription, protein translation, ribosome synthesis, and autophagy (*Loewith and Hall, 2011*). A characteristic that distinguishes TORC1 is its interaction with KOG1 (raptor in mammals) and its sensitivity to the rapamycin-FKBP12 complex (*Adami et al., 2007*). The Tor kinase is widely conserved both in terms of its structural characteristics and its role as the target of Fkbp-rapamycin (*Tatebe and Shiozaki, 2017*). Typically, the Tor kinase exhibits similar structural characteristics, including the HEAT-like, FAT, FRB, kinase, and FATC domains (*Tatebe and Shiozaki, 2017*). The FRB domain is the binding region of the Fkbp-rapamycin complex, and rapamycin resistance is usually caused by the disruption of the FRB domain (*Loewith and Hall, 2011*). Homology analysis and phylogenetic studies have identified a single predicted *torA* gene within the *A. flavus* genome. To elucidate the functional role of *torA* in *A. flavus*, our initial approach involved an attempt to generate an *torA* deletion strain through homologous recombination. However, these efforts were unsuccessful. We speculated that the deletion of the *torA* gene may be lethal in *A. flavus*. This observation aligns with the situation observed in *F. graminearum* (*Yu et al., 2014*). Therefore, to investigate the function of *torA* in *A. flavus*, we constructed a mutant strain (*xylPtorA*) using an xylose-inducible promoter, which allows for conditional induction with the addition of xylose. We observed that the *xylPtorA* mutant is completely unable to grow in YGT medium without xylose, whereas partial growth is restored in YXT medium with the addition of xylose (*Figure 3A*). These results indicated that TorA is essential for the growth of *A. flavus*. We also discovered that the TorA kinase is involved in the vegetative growth, asexual development, sclerotial, and aflatoxin production of *A. flavus* (*Figure 3A and B*, *Figure 3—figure supplement 2*). In addition, the the *xylPtorA* mutant shows a high sensitivity to rapamycin and SDS (*Figure 3A*). Our findings have demonstrated that the TorA kinase plays a crucial role in the processes of morphogenesis, aflatoxin biosynthesis, and multiple stress responses in *A. flavus*. Previous findings demonstrated that the Tor kinase could function as a molecular switch connecting an iron cue to defend against pathogen infection in *C. elegans* (*Ma et al., 2021*). In *A. fumigatus*, the Tor kinase represents a central regulatory node controlling genes and proteins involved in nutrient sensing, stress response, cell cycle progression, protein biosynthesis, and degradation, as well as adaptation to low iron conditions. In summary, the Tor kinase acts as a central regulator in the TOR signaling pathway, playing a crucial role in the regulation of the cell cycle, protein synthesis, and cellular energy metabolism. Its influence on key cellular processes makes it a promising target for future research to control fungal growth and aflatoxin production.

The serine/threonine protein kinase Sch9, a critical downstream effector of TORC1 in *S. cerevisiae*, has been extensively studied for its roles in stress resistance, longevity, and nutrient sensing (*Pascual-Ahuir and Proft, 2007*). In addition, the TORC1-Sch9 pathway is also recognized as crucial mediator of chronological lifespan (*Pascual-Ahuir and Proft, 2007*). In filamentous fungi, an intricate network of interactions exists between the TOR and HOG pathways. In *F. graminearum* (*Gu et al., 2015*), FgSch9 and FgHog1 mutants displayed increased susceptibility to osmotic and oxidative stresses. The phosphorylation level of FgHog1 increased significantly in FgSch9 mutant. Additionally, the affinity capture-MS assay showed that the Tor kinase was among the proteins co-purifying with FgSch9. These results suggested that FgSch9 serves as an intermediary for the TOR and HOG signaling pathways (*Gu et al., 2015*). Furthermore, FgSch9 plays a crucial role in the regulation of various biological processes such as hyphal differentiation, asexual development, virulence, and DON biosynthesis (*Gu et al., 2015*). In *A. fumigatus*, the SchA mutant was sensitive to rapamycin, high concentrations of calcium and hyperosmotic stress, and SchA was involved in iron metabolism. The SchA mutant exhibited enhanced phosphorylation of SakA in response to osmotic stress (*Alves de Castro and Dos Reis, 2016*). In this study, we discovered that Sch9 is involved in regulating aflatoxin biosynthesis (*Figure 4C*), and responses to osmotic and rapamycin stress (*Figure 4B and D*). The S_TKc domain

deletion strain (sch9[ΔS_TKc]), S_TK_X domain deletion strain (sch9[ΔS_TK_X]), and sch9[K340A] mutant displayed a similar phenotype to the Δsch9 strain (*Figure 4B–D*). These findings revealed that Sch9 plays a significant role in the biological synthesis of aflatoxin and in responding to various stresses. This role is primarily mediated through its kinase domain, kinase C domain, and ATP binding site K340. We also found that the phosphorylation level of Hog1 increased significantly in the Δsch9 strain under osmotic stress (*Figure 4E*). Based on the above results, we hypothesized that Sch9 was involved in the osmotic stress response by modulating the HOG pathway in *A. flavus*.

The phosphorylation state of Tap42 is crucial for its interaction with PP2A subunits and is regulated by a balance between Tor-mediated phosphorylation and Cdc55/Tpd3-mediated dephosphorylation in yeast (*Jiang and Broach, 1999*). The genome of *A. flavus* contains a homolog of *tap42* named *tapA* (AFLA_092770), which is predicted to encode a 355 amino acid protein (sharing 31.9% identity with *S. cerevisiae* Tap42). To further determine the function of the TapA regulator in the TOR signaling pathway of *A. flavus*, we conducted targeted gene deletion experiments on *tapA*. We failed to obtain a null mutant, we hypothesized that *tapA* knockout was lethal in *A. flavus*. To investigate the function of TapA, we employed homologous recombination, to replace the native promoter with the strong *gpdA* promoter, thereby constructing an overexpression strain (*OE::tapA*) with the enhanced promoter. Compared with WT strain, the *OE::tapA* strain were more sensitive to rapamycin and SDS stress, indicating that *tapA* gene plays an important role in regulating cell wall stress (*Figure 5B and F*). In *S. pombe*, the protein Tip41, which interacts with Tap42, plays a significant role in the cellular responses to nitrogen sources by regulating type 2A phosphatases (*Fenyvuesvolgyi et al., 2005*). In *S. cerevisiae*, Tap42-phosphatase complexes associate with TORC1, whereas Tip41 functions to attenuate TORC1 signaling by stimulating the PP2A phosphatase (*Jacinto et al., 2001*). Unlike its yeast counterpart Tip41, TIPRL, a mammalian Tip41-like protein, exerts a positive influence on mTORC1 signaling by interacting with PP2Ac (*Nakashima et al., 2013*). While some functions of Tap42 and its binding partners Tip41 may be conserved, the interaction between Tip41 and Tap42 was not observed in the filamentous fungi. The yeast two-hybrid assay showed that FgTip41 interacts with the phosphatase FgPpg1 rather than FgTap42, while the FgTap42 bind with phosphatase FgPp2A, implying that FgTap42, FgPpg1, and FgTip41 form a heterotrimer in *F. graminearum* (*Yu et al., 2014*). In the rice blast fungus *M. oryzae*, MoTip41 does not interact with MoTap42. However, MoTip41 can bind with the phosphatase MoPpe1, thereby facilitating crosstalk between the TOR pathway and the CWI pathway (*Qian et al., 2021*). In *A. flavus*, the *OE::tapA* and Δ*tipA* strains showed increased sensitivity to cell wall stress compared with the WT strain (*Figures 5B and 6H*). Based on the conserved roles of Tap42 and Tip41 in yeast and other filamentous fungi, we hypothesized that TapA and TipA in *A. flavus* also participate in CWI pathway. In summary, our study provided important insights into the role of TapA and TipA in the TOR signaling pathway of *A. flavus*, particularly in relation to cell wall stress. However, their interaction and implications for TOR and CWI signaling in *A. flavus* remain to be elucidated.

In *S. cerevisiae*, Tap42 can bind and regulate various PP2A phosphatases, including Pph3, Pph21, Pph22, Sit4, and Ppg1. This interaction plays a crucial role in the TOR signaling pathway, particularly in response to rapamycin treatment or nutrient deprivation (*Jacinto et al., 2001*). These phosphatases play important roles in other fungi and are involved in a variety of cellular processes, including growth, morphogenesis, and virulence. For example, the phosphatase Sit4 plays a crucial role in regulating cell growth, morphogenesis, and virulence in *C. albicans* (*Han et al., 2019*). Similarly, FgSit4 and FgPpg1 are associated with various cellular processes, including mycelial growth, conidiation, DON biosynthesis, and virulence in *F. graminearum* (*Yu et al., 2014*). These findings indicated the conserved role of PP2A phosphatases in fungal biology. In addition to their roles in growth and development, FgSit4 and FgPpg1 are involved in the regulation of various signaling pathways, such as the CWI pathway and LDs biogenesis regulation pathway (*Yu et al., 2014*). In this study, we found that the phosphatases SitA and Ppg1 play critical role in vegetative growth, conidiation, sclerotia formation, and aflatoxin biosynthesis in *A. flavus* (*Figure 7A–C*). Additionally, the phosphatases SitA and Ppg1 also affect pathogenicity and LD biogenesis (*Figures 8A, 10C and D*). This highlights the multifaceted influence of PP2A phosphatases on cellular signaling.

After the deletion of the *sitA* and *ppg1* genes, the expression level of genes related to cell wall synthesis has decreased significantly, resulting in damage to the cell wall and showing a higher sensitivity to CWI stress (*Figure 9C*). Western blotting results showed that the phosphorylation level of

MAPK kinase Slt2 has increased significantly in the Δ*sitA* and Δ*ppg1* strains under cell wall stress, indicating that SitA and Ppg1 exert a negative regulatory effect on the CWI pathway (*Figure 9D*). In *M. oryzae*, MoPpe1 and MoSap1 serve as an adaptor complex that connects the CWI and TOR signaling pathways. The activation of the TOR pathway leads to the inhibition of the CWI pathway, highlighting a critical regulatory interplay between these two pathways in fungal biology (*Qian et al., 2018*). We speculated that SitA and Ppg1 might exhibit similar regulation within the CWI and TOR pathways across different fungal species. In addition, the phosphatase MoPpe1 interacts with the phosphatase-associated protein MoTip41, facilitating the exchange of information between the TOR and CWI signaling pathways (*Qian et al., 2018*). These results suggested that the phosphatase Ppe1 engages in interactions with various proteins to effectively coordinate the TOR and CWI signaling pathways, thereby regulating the growth and pathogenicity of rice blast fungus *M. oryzae* (*Qian et al., 2018*). Given the conserved functions of PP2A phosphatases and the observed roles of SitA, Ppg1, TapA, and TipA in cell wall stress response, we hypothesized that TapA-phosphatase complexes may collaborate to regulate CWI processes in *A. flavus*. This hypothesis is supported by the similar functions exhibited by these proteins.

In *F. graminearum*, the phosphatase complex FgNem1/Spo7-FgPah1 cascade plays significant roles in fungal development, DON production, and LD biogenesis (*Liu et al., 2019*). Our study explored the conservation and functional significance of Nem1 and Spo7 homologs in *A. flavus*, revealing their pivotal roles in vegetative growth and secondary metabolism (*Figure 11A and B*). Our results indicated that LD biogenesis is facilitated by the Nem1/Spo7 and SitA/Ppg1 pathways in *A. flavus* (*Figures 10C, D and 11E*). These observations suggested a conserved and central role for PP2A phosphatases as core regulators within the LD biogenesis signaling network, highlighting their importance in cellular lipid metabolism and energy storage. While our findings underscored the significance of phosphatases in LD biogenesis, it is also plausible that other kinases contribute to the regulation of this process. To gain a comprehensive understanding of the precise molecular mechanism, it is essential to investigate the interplay between phosphatases and other kinases.

In conclusion, we identified the complex regulatory network that provides important insight into the interplay between the TOR, HOG, and CWI signaling pathways, and how they collectively regulate the development, aflatoxin biosynthesis, and pathogenicity of *A. flavus*. By elucidating the functions of genes associated with the TOR signaling pathway, our research could establish a solid foundation for comprehending the molecular mechanism underlying the aflatoxin biosynthesis of *A. flavus*. Our results could also contribute to the development of targeted strategies to manage aflatoxin contamination and mitigate the impact of *A. flavus* infections.

## Materials and methods
### Strains and cultural conditions
The *A. flavus* strains in this study were shown in *Supplementary file 1a*. All strains were cultivated on four different types of agar media: PDA, YES, YGT, and YXT. The growth and conidiation assays were conducted at 37°C. Additionally, aflatoxin production was assessed using YES and YXT liquid medium at 29°C.

### Constructions of gene deletion and complemented mutants
The construction of gene deletion and complemented strains was carried out using the SOE-PCR strategy, following previously established protocols (*Tan et al., 2013*). To generate the $^{xylP}torA$ mutant of *A. flavus*, an *torA* xylose promoter mutant cassette was synthesized via overlapping extension PCR. This cassette was designed to incorporate an upstream fragment of the *torA* gene, a selectable marker gene (*pyrG* from *A. fumigatus*), the xylP xylose-inducible promoter (*Zadra et al., 2000*), and a fragment of the *torA* coding sequence. Subsequently, this composite cassette was integrated into the *A. flavus* genome via homologous recombination, specifically replacing the native *torA* promoter with the inducible xylose promoter. To construct overexpression cassettes for the *tapA* gene, we utilized a molecular cloning strategy that entailed the fusion of PCR-amplified fragments from the *pyrG*-selectable marker and the *gpdA* promoter. Subsequently, employing a homologous recombination approach, we inserted this composite fragment upstream of the *tapA* gene, effectively replacing its native promoter with the stronger *gpdA* promoter. The primers utilized for the amplification of the

flanking sequences of each gene can be found in **Supplementary file 1b**. The putative gene deletion mutants were validated through PCR assays using the appropriate primers (**Figure 2—figure supplement 1**), and their confirmation was subsequently obtained through sequencing analysis.

## Growth, conidiation, and sclerotia analysis

To evaluate mycelial growth and conidia formation, 1 μL of $10^6$ spores/mL conidia of all strains was point inoculated onto the center of the PDA, YGT, or YXT medium, and then cultured at 37°C for 5 days in the dark. The conidia were washed with 0.07% Tween-20 and counted using a hemocytometer and microscope (**Ren et al., 2018**). For sclerotia formation analysis, each strain was inoculated and cultivated on CM and YXT agar medium at 37°C in the dark for 7 days, and then 75% ethanol was used to wash away mycelia and conidia on the surface of the medium (**Nie et al., 2016**). These assays were repeated at least three times.

## Extraction and quantitative analysis of aflatoxin

To extract and quantify aflatoxins production, 1 μL of $10^6$ spores/mL conidia of all strains was point inoculated onto liquid YES medium, and cultures were then incubated at 29°C in the dark for 6 days. Aflatoxins were extracted from the media and detected using TLC (**Nie et al., 2016**). In addition, the standard $AFB_1$ was employed for visual comparison, and aflatoxins were quantitatively analyzed using the ImageJ tool. The assays were repeated at least three times.

## Stress assay

To determine the roles of the gene in *A. flavus* responding to various environmental stresses, 1 μL of $10^6$ spores/mL conidia of all strains were point inoculated onto these PDA, YGT, YXT, or YES solid plates supplemented with various agents. These agents included 1 M NaCl, 1 M KCl, and 1 M $CaCl_2$ for inducing osmotic stress, as well as 100 μg/mL or 300 μg/mL SDS, 200 μg/mL CFW, and 200 μg/mL CR for inducing cell wall stress. After 5 days of incubation at 37°C in the dark, the relative rates of inhibition were determined (**Tan et al., 2021**). The assays were repeated at least three times.

## Seeds infection assays

To measure the pathogenicity of *A. flavus*, all peanut and maize seeds were washed in 0.05% sodium hypochlorite, 75% ethanol, and sterile water, respectively. Subsequently, all peanut and maize seeds were immersed in sterile water containing $10^6$ spores/mL of all strains at 29°C in the dark for 60 min. After the conidia adhesion, the seeds were cultured at 29°C for 6 days in the dark (**Tan et al., 2021**).

## Quantitative RT-PCR

For the real-time quantitative PCR (qPCR) assays, the *A. flavus* strains were cultivated in the PDA medium at 29°C for 3 days in the dark. Total RNA was extracted from the mycelia of all strains, and cDNA was synthesized with the First Strand cDNA Synthesis Kit, and then the cDNA was used as the template for qRT-PCR analysis with SYBR Green qPCR mix. The relative transcript level of each gene was calculated by the $2^{-\Delta\Delta CT}$ method (**Livak and Schmittgen, 2001**).

## Western blot analysis

To extract the total protein, the mycelia of each strain were cultivated in YES liquid medium for 48 hr, then the sample was grinded under liquid nitrogen to a fine powder and transferred to a 1.5 mL microcentrifuge tube added with RIPA lysis buffer and 1 mM PMSF protease inhibitor. Proteins were separated by 12% SDS-PAGE and transferred onto a polyvinylidene fluoride membrane. Phosphorylation levels of Hog1 and Slt2 were detected by Phospho-p38 MAPK antibody (Cell Signaling Technology, Boston, MA, USA) and Phospho-p44/42 MAPK (Erk 1/2) antibody (Cell Signaling Technology, MA, USA), respectively. Monoclonal antibody Hog1 (Santa Cruz Biotechnology, Dallas, TX, USA) and Mpk1 antibody (Santa Cruz Biotechnology, Dallas, TX, USA) were used as loading controls. Total Slt2 levels were quantified with an anti-Mpk1 antibody. The experiment was conducted three times independently.

## Statistical analysis

All data were measured with means ± standard deviation (SD) of three biological replicates. GraphPad Prism 7 software was used for conducting statistical and significance analysis, and

p-values<0.05 represented statistical significance. Student's t-test was used to compare the two means for differences, whereas Tukey's multiple comparisons were used for testing multiple comparisons.

## Acknowledgements

The authors also thank Zhenhong Zhuang, Jun Yuan, Yu Wang, Kunzhi Jia, Xiuna Wang, and Xinyi Nie for their help in the experiments.

## Additional information

### Funding

| Funder | Grant reference number | Author |
|---|---|---|
| National Natural Science Foundation of China | Grant No. 31972214 | Shihua Wang |

The funders had no role in study design, data collection and interpretation, or the decision to submit the work for publication.

### Author contributions

Guoqi Li, Data curation, Software, Formal analysis, Supervision, Validation, Investigation, Methodology, Writing - original draft, Project administration, Writing - review and editing; Xiaohong Cao, Data curation, Investigation, Methodology, Project administration; Elisabeth Tumukunde, Writing - review and editing; Qianhua Zeng, Investigation, Project administration; Shihua Wang, Funding acquisition, Writing - review and editing

### Author ORCIDs

Guoqi Li ⓘ http://orcid.org/0009-0005-6512-0892
Shihua Wang ⓘ http://orcid.org/0000-0002-1544-5972

Reviewer #1 (Public Review): https://doi.org/10.7554/eLife.89478.5.sa1
Reviewer #2 (Public Review): https://doi.org/10.7554/eLife.89478.5.sa2
Author response https://doi.org/10.7554/eLife.89478.5.sa3

## Additional files

### Supplementary files
• Supplementary file 1. Comprehensive list of strains and primers for this study.
• MDAR checklist

### Data availability
The authors confirm that the data supporting the findings of this study are available within the article and supplementary file. Source data files have been provided for *Figures 4 and 9* and *Figure 2— figure supplement 1*.

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
