## [Editor Report · eLife assessment]

This manuscript provides **important** information about the influence of TOR signaling pathway on development and aflatoxin production in the plant and human fungal pathogen Aspergillus flavus. Compared to an earlier version, the authors have addressed most of the concerns of the reviewers, including the **convincing** demonstration of the essential TOR pathway in this fungus by constructing a xylose promoter mutant strain.

---

## [Referee Report · Reviewer #1 (Public Review)]

This paper reports the useful discovery of the roles and signaling components of the TOR pathway in vegetative growth, sexual development, stress response, and aflatoxin production in Aspergillus flavus.

While I acknowledge the authors' effort in conducting Southern blot analysis to address my prior concern regarding the presence of dual copies of torA and tapA, I find their current resolution inadequate. Specifically, the simple deletion of the respective result sections for torA and tapA significantly impacts the overall significance of this study. The repeated unsuccessful attempts to generate correct mutants only offer circumstantial evidence, as technical issues may have been a contributing factor. Therefore, instead of merely removing these sections, it is essential for the authors to present more compelling experimental data demonstrating that torA and tapA are indeed vital for the viability of A. flavus. Such data would enhance the overall significance of this study.

---

## [Referee Report · Reviewer #2 (Public Review)]

In this study, authors identified TOR, HOG and CWI signaling network genes as modulators of the development, aflatoxin biosynthesis and pathogenicity of A. flavus by gene deletions combined with phenotypic observation. They also analyzed the specific regulatory process and proposed that the TOR signaling pathway interacts with other signaling pathways (MAPK, CWI, calcineurin-CrzA pathway) to regulate the responses to various environmental stresses. Notably, they found that FKBP3 is involved in sclerotia and aflatoxin biosynthesis and rapamycin resistance in A. flavus, especially that the conserved site K19 of FKBP3 plays a key role in regulating aflatoxin biosynthesis. In general, the study involved a heavy workload and the findings are potentially interesting and important for understanding or controlling the aflatoxin biosynthesis. However, the findings have not been deeply explored and the conclusions mostly are based on parallel phenotypic observations.

---

## [Author Response]

The following is the authors’ response to the previous reviews.

**Reviewer #1 (Public Review):**
While I acknowledge the authors' effort in conducting Southern blot analysis to address my prior concern regarding the presence of dual copies of torA and tapA, I find their current resolution inadequate. Specifically, the simple deletion of the respective result sections for torA and tapA significantly impacts the overall significance of this study. The repeated unsuccessful attempts to generate correct mutants only offer circumstantial evidence, as technical issues may have been a contributing factor. Therefore, instead of merely removing these sections, it is essential for the authors to present more compelling experimental data demonstrating that torA and tapA are indeed vital for the viability of A. flavus. Such data would enhance the overall significance of this study.

We agree and appreciate reviewer's important comments on our manuscript. In this version, we address this issue by providing additional experimental data to further support the importance of *torA* and *tapA* in the viability of *A. flavus*. We conducted additional experiments to generate more compelling evidence regarding the essential role of *torA* and *tapA* in the growth and development of *A. flavus*. We constructed a mutant strain (*xylPtorA*) using an xylose-inducible promoter, which allows for conditional induction with the addition of xylose (Lines 204-238, page 10).

Due to the unsuccessful construction of TapA knockout strains and xylose promoter replacement strains, we used homologous recombination to replace the original promoter with the *gpdA* strong promoter for overexpression of *tapA* (*OE*::*tapA*). We thank reviewer for highlighting this important aspect, and we revise our manuscript accordingly to enhance its overall significance (Lines 277-297, page 13). We are grateful for the opportunity to enhance our manuscript and believe these revisions provide a more comprehensive understanding of the roles of *torA* and *tapA* in *A. flavus*.

**Reviewer #1 (Recommendations For The Authors):**
Minor commentsLines 421-423 and 465-466: these sentences are grammatically awkward. Please rephrase them.

Thank you for your feedback on our manuscript. We conducted additional experiments, so we have removed the sentence from the manuscript to maintain coherence and avoid redundancy.

**Reviewer #2 (Public Review):**
In this study, authors identified TOR, HOG and CWI signaling network genes as modulators of the development, aflatoxin biosynthesis and pathogenicity of *A. flavus* by gene deletions combined with phenotypic observation. They also analyzed the specific regulatory process and proposed that the TOR signaling pathway interacts with other signaling pathways (MAPK, CWI, calcineurin-CrzA pathway) to regulate the responses to various environmental stresses. Notably, they found that FKBP3 is involved in sclerotia and aflatoxin biosynthesis and rapamycin resistance in *A. flavus*, especially that the conserved site K19 of FKBP3 plays a key role in regulating aflatoxin biosynthesis. In general, the study involved a heavy workload and the findings are potentially interesting and important for understanding or controlling the aflatoxin biosynthesis. However, the findings have not been deeply explored and the conclusions mostly are based on parallel phenotypic observations.

Thank you for your constructive comments on our manuscript. In response to your comments, we have conducted additional experiments, including the construction of a xylose promoter mutant strain and an overexpression strain. We have also expanded the discussion section to provide a more comprehensive analysis of our findings in the context of existing literature. Thank you again for your insightful feedback, which has been instrumental in improving the quality of our work. (Lines 464-469, page 22).

**Reviewer #2 (Recommendations For The Authors):**
Point 1: Our findings revealed that both the tor and tapA genes are present in double copies in our strains, which guided our decision to construct single-copy deletion strains using homologous recombination However, the tor gene in A. flavus exhibited varying copy numbers, as was confirmed by absolute quantification PCR at the genome level (Table S1). However, it is hard to understand for Table S1: Estimation of copy number of tor gene in A. flavus toro and sumoo stand for the initial copy number, and the data are graphed as the mean {plus minus} 95%confidence limit. CN is copy number. As indicated in the Methods, Using sumo gene as reference, the tor and tapA gene copy number was calculated by standard curve. In Table S1 of WT, for tor gene, CN value is1412537 compared to 1698243 in tor+/-, for the reference gene sumo,794328 compared to1584893, how these data could support copy gene numbers of tor?

Thank you for your insightful comments. We understand the confusion with the data presented in Table S1 regarding the copy number estimation of the *torA* gene in *A. flavus*. We apologize for not providing a clear explanation for the data in the table. Quantitative real-time PCR (qPCR) is widely used to determine the copy number of a specific gene. It involves amplifying the gene of interest and a reference gene simultaneously using specific primers and probes. By comparing the amplification curves of the gene of interest and the reference gene, we can estimate the relative copy number of the gene.

To address your concern and provide more accurate information, we have re-performed the copy number analysis using southern blot. Southern blot analysis allows for the direct estimation of gene copy number by hybridizing genomic DNA with a specific probe for the gene. This method provides more reliable and accurate results in determining gene copy numbers. We discovered that the *A. flavus* genome contains a single copy of the *torA* gene. Consequently, we conducted additional experiments to elucidate its function. Specifically, we generated strains with a xylose-inducible promoter system to modulate the expression of *torA* (Lines 204-238, page 10).

Point 2: In response: For the knockout of the FRB domain, we used the homologous recombination method, but because tor genes are double-copy genes, there are also double copies in the FRB domain. Despite our efforts, we encountered challenges in precisely determining the location of the other copy of the tor gene. I could not understand these consistent data, why not for using sequencing?

Thank you for your valuable feedback. We determined again and confirmed that the *torA* gene is a single copy. So we removed this part of the results to avoid any ambiguity or potential misinterpretation.

Point 3: Response in Due to the large number of genes involved, we did not perform a complementation experiment. If there were no complementation data, how to demonstrate data are solid?

Thank you for your important suggestion. We understand that complementation experiments are commonly used to validate gene deletions. Therefore, to ensure the reliability of our data, we have conducted supplementary experiments on specific gene deletions, such as Δ*sitA*-C and Δ*ppg1*-C. Thank you again for your positive comments and valuable suggestions, which have significantly contributed to enhancing the quality of our manuscript (Lines 320-322, page 15).

Point 4: Acknowledge the confusion? We acknowledge the confusion in our presentation and will ensure that accurate genetic nomenclature is used consistently

Thank you for your comments on our manuscript. We recognize the importance of precise and consistent use of genetic nomenclature, as it is critical for the clarity and integrity of our research findings. We have carefully reviewed the sections of our manuscript where genetic terms were used and have made the necessary corrections to ensure that all nomenclature is accurate and used consistently throughout the text.

Point 5: In the revised version of new manuscript, southern blotting was carried out and found only one copy was existed for tested genes at last. Thus, whole manuscript conclusions should be changed. In addition, Reviewer 1 suggestion for using Illumina-sequence strategy, their tor and tapA mutants could be verified whether they are aneuploid?

We would like to express our gratitude for your insightful comments and suggestions. Following the new experimental data obtained from Southern blotting, we have identified that only one copy of the tested genes exists, and we have revised our conclusions throughout the manuscript. This has led to a significant reinterpretation of our results and a reassessment of the implications for our study. Based on this result, we designed and constructed strains with the tor gene under the control of a xylose-inducible promoter. This approach allows for the conditional expression of the tor gene. Thank you once again for your meticulous review (Lines 204-238, page 10).